# Alloying and confinement effects on hierarchically nanoporous CuAu for efficient electrocatalytic semi-hydrogenation of terminal alkynes

Linghu Meng[1,5], Cheng-Wei Kao[2,5], Zhen Wang[1,5], Jun Ma[3], Peifeng Huang[3], Nan Zhao[4], Xin Zheng[4], Ming Peng[1], Ying-Rui Lu [2] & Yongwen Tan [1] ✉

Electrocatalytic alkynes semi-hydrogenation to produce alkenes with high yield and Faradaic efficiency remains technically challenging because of kinetically favorable hydrogen evolution reaction and over-hydrogenation. Here, we propose a hierarchically nanoporous $Cu_{50}Au_{50}$ alloy to improve electrocatalytic performance toward semi-hydrogenation of alkynes. Using Operando X-ray absorption spectroscopy and density functional theory calculations, we find that Au modulate the electronic structure of Cu, which could intrinsically inhibit the combination of H* to form $H_2$ and weaken alkene adsorption, thus promoting alkyne semi-hydrogenation and hampering alkene over-hydrogenation. Finite element method simulations and experimental results unveil that hierarchically nanoporous catalysts induce a local microenvironment with abundant $K^+$ cations by enhancing the electric field within the nanopore, accelerating water electrolysis to form more H*, thereby promoting the conversion of alkynes. As a result, the nanoporous $Cu_{50}Au_{50}$ electrocatalyst achieves highly efficient electrocatalytic semi-hydrogenation of alkynes with 94% conversion, 100% selectivity, and a 92% Faradaic efficiency over wide potential window. This work provides a general guidance of the rational design for high-performance electrocatalytic transfer semi-hydrogenation catalysts.

Selective semi-hydrogenation of alkynes into alkenes is an important catalytic reaction for the synthesis of value-added fine chemicals[1–3]. Currently, the semi-hydrogenation reaction of alkynes is commonly operated (thermo)chemical hydrogenation (TCH), where the flammable and explosive $H_2$ gas as the source is catalytically dissociated for subsequent hydrogenation of alkynes under high temperature and pressure conditions[4–6]. However, a large part of the gaseous $H_2$ source

of the TCH processes is produced by the fossil fuel-based steam reforming process that inevitably results in high energy consumption and releases massive amounts of $CO_2$[7]. Alternatively, electrochemical hydrogenation technology, powered by renewable and clean energy, which would be more attractive and sustainable due to its low cost, high safety and environment friendly[8]. In this regard, a large amount of surface active hydrogen (H*) generated by water electrolysis can

[1]College of Materials Science and Engineering, State Key Laboratory of Advanced Design and Manufacturing Technology for Vehicle Body, Hunan University, Changsha 410082 Hunan, China. [2]National Synchrotron Radiation Research Center, Hsinchu 300092, Taiwan. [3]College of Mechanical and Vehicle Engineering, Hunan University, Changsha 410082 Hunan, China. [4]Electrical Power Research Institute of Yunnan Power Grid Co. Ltd, North China Electric Power, Kunming 650217 Yunnan, China. [5]These authors contributed equally: Linghu Meng, Cheng-Wei Kao, Zhen Wang. ✉e-mail: tanyw@hnu.edu.cn

directly utilized as a sustainable hydrogen source for electrocatalytic transfer hydrogenation to avoid the use of hazardous hydrogen[9]. Unfortunately, electrochemical hydrogenation reaction process, often accompany with the competitive hydrogen evolution reaction (HER) due to the favorable water dissociation, eventually leading to poor Faradaic efficiency (FE) and conversions[10,11]. Additionally, the lack of effective control on the specific adsorption of intermediates at high production conversions greatly limits the yields of targeted products due to over-hydrogenation reactions[12]. Therefore, it is highly desirable to develop an efficient electrocatalyst to concurrently realize the electrocatalytic semi-hydrogenation of alkynes with high alkyne conversion rate, high alkene selectivity and Faradaic efficiency, which are not readily available to access by the present methods, especially at gram-scale[13].

Earth-abundant Cu-based nanomaterials have been widely exploited as high-efficiency catalysts for organic semi-hydrogenation due to their strong capabilities of electron donation and activation of hydrogenation steps[11,14-17]. However, the strong adsorption of alkene intermediates on Cu-based catalysts makes them susceptible to over-hydrogenation, leading to poor selectivity[18]. Fortunately, the electronic/geometric configurations of Cu-based catalysts could usually be regulated by the incorporation of metallic or non-metallic species to optimize the adsorption of substrates and reaction intermediates on the catalyst surface, thereby influencing the activity and selectivity[18-22]. Despite great improvements, the electrocatalytic semi-hydrogenation of alkynes performance of Cu-based nanostructures is still unsatisfied due to the weak hydrogenation ability of Cu itself. Therefore, the limited selectivity and productivity are achieved to drive the alkynes semi-hydrogenation with only 3.6 to 10% FEs because of insufficient capability of H* formation on mono-metallic surface[13]. In this regard, the activation of water at the electrode-electrolyte interface is crucial for the generation and utilization of active H* in electrocatalytic reduction reactions using water as a hydrogen source[23]. An alternative approach is to create a nanoconfinement microenvironment in porous structure, whereby a high local electric field that increases the concentration of reactants or electrolytes and the moderation of active H* adsorption around the nanopore, thus facilitating the target reaction[24-27]. Nevertheless, well-designed nanoporous catalysts and their intrinsic mechanism for electrochemical selective semi-hydrogenation still rare at present. Therefore, it is of great significance to construct nanoporous structures, which can not only expose more active sites and accelerate electron/mass transfers[28], but also exhibit a high local electric field that increases the concentration of reactants or electrolytes around the nanopore, hence enhancing electrochemical selective semi-hydrogenation performance of alkynes.

Herein, we present a hierarchically nanoporous $Cu_{50}Au_{50}$ (Hnp-$Cu_{50}Au_{50}$) alloy with interconnected macroporous channels and numerous nanopores, which simultaneously enhance the selectivity and reactivity for electrochemical alkynes semi-hydrogenation by combining the alloying effect and confinement effect. Operando X-ray absorption spectroscopy (XAS), in situ Raman spectroscopy and density functional theory (DFT) unravel that the introduction of Au into the Cu lattice could optimize the electronic structure to prevent the combination of H* to form $H_2$ and weaken the adsorption and interaction capacity of alkenes to inhibit over-hydrogenation, thus promoting semi-hydrogenation selectivity of alkenes production. Finite element method simulations reveal that hierarchically nanoporous structure enable the accumulated $K^+$ by enhancing the local electric field within the nanopores, accelerating $H_2O$ electrolysis to form more H*, and improve semi-hydrogenation performance of alkynes. As a result, the high phenylacetylene semi-hydrogenation performance with 94% conversion, 100% selectivity, and a 92% Faradaic efficiency over Hnp-$Cu_{50}Au_{50}$ alloy is demonstrated for the production of styrene. Importantly, a gram-scale synthetic process

with high FE of 68% is achieved at a current density of 25 mA cm$^{-2}$ for a continuous styrene production, demonstrating the great potential of this electrochemical selective semi-hydrogenation process for replacing conventional TCH processes.

## Results

### Materials synthesis and characterizations

The hierarchically nanoporous $Cu_{50}Au_{50}$ alloy with bimodal ligament and pore size distribution was prepared by two-step chemical dealloying from a $Al_{80}Cu_{18}Au_2$ precursor alloy (Supplementary Figs. 1 and 2). Typically, the α-Al phase and a small percentage of Al existing in Al-Cu-Au intermetallic phase in $Al_{80}Cu_{18}Au_2$ precursor alloy were selectively etched after first-step dealloying in KOH aqueous solution, leaving first ordered porous structure with an average size of ~80 nm (Supplementary Fig. 3). Subsequently hierarchically nanoporous CuAu alloys with different Cu/Au ratio were obtained by controlling corrosion time in $HNO_3$ solutions (Supplementary Fig. 4), thereby leading to a hierarchical CuAu structure with interconnected macroporous channels with a size of 80 nm and highly uniformly nanopores with a size of 5 nm (Fig. 1a). For comparison, unimodal nanoporous Cu (np-Cu), nanoporous Au (np-Au), and nanoporous $Cu_{50}Au_{50}$ (np-$Cu_{50}Au_{50}$) were also prepared via one-step dealloying approach (Supplementary Fig. 5). The representative high-angle annular dark field scanning transmission electron microscopy (HAADF-STEM) image of Hnp-$Cu_{50}Au_{50}$ alloy displays that secondary ligaments consist of interconnected crystalline features with a diameter of ~5 nm (Fig. 1b), the uniform mesopores could generate a large active surface area that enables abundant active sites for electrocatalytic reactions. As shown in the X-ray diffraction (XRD) patterns (Supplementary Fig. 4), the hierarchically nanoporous CuAu alloys with different Cu/Au ratios retain the face-centered cubic (FCC) structure, which is matched with mono-metallic Cu and Au. However, the lattice spacing of the hierarchically nanoporous CuAu alloys depend on Cu/Au ratios and decrease with the increase of Au fractions due to the substitution of Cu by Au with a greater atomic radius[29]. The measured d-spacing of Hnp-$Cu_{50}Au_{50}$ alloy is 0.218 nm (Fig. 1c), which can be indexed as (111) plane of the CuAu alloy[30]. The STEM-energy-dispersive spectrometry (STEM-EDS) mapping images reveal the uniform distribution of Cu and Au element throughout the whole nanoporous networks with the atomic ratio of 50:50 (Fig. 1d). The alloying effect in catalysis often arises from the modification of the valence state and coordination environment of active atoms. X-ray photoelectron spectroscopy (XPS) reveals the addition of Au leads to the negative shift of Cu 2p binding energy (Fig. 1e), while the binding energy of Au 4f increases by 0.30 eV for Hnp-$Cu_{50}Au_{50}$ alloy (Supplementary Fig. 6). These core-level shifts are caused by the valence state changes due to the hybridization between neighboring atoms upon alloying, which results in the electron transfer from Au to Cu[31]. By tracking changes in X-ray absorption near-edge structure (XANES) and extended X-ray absorption fine structure (EXAFS) spectra, we further examined the local electronic and coordination structure of Cu and Au. The Cu K-edge XANES spectrum of Hnp-$Cu_{50}Au_{50}$ shows an obvious shift towards lower energy compared to np-Cu and Cu foil (Fig. 1f), while the Au L3-edge white-line slightly increases compared to np-Au and Au foil (Supplementary Fig. 7), indicating electron transfer from Au to Cu, which is consistent with XPS spectra results[32]. Figure 1g presents the Fourier-transformed Cu K-edge EXAFS spectral. In the R space, Hnp-$Cu_{50}Au_{50}$ exhibits a prominent peak at ~1.5 Å from the Cu-O bonds due to the dealloying process or be oxidized by air. The peak at 2.26 Å is different from the Cu-Cu characteristic peak (~2.22 Å) of np-Cu and Cu foil due to the formation of Cu-Au bonds after incorporation of Au into the Cu lattice[33,34], which is further confirmed by Au L3-edge EXAFS spectrum (Supplementary Fig. 7). Importantly, the FT-EXAFS spectrum for Hnp-$Cu_{50}Au_{50}$ shows the much lower Cu-Au peak intensity in comparison with Cu foil, resulting from the grains become smaller, which could

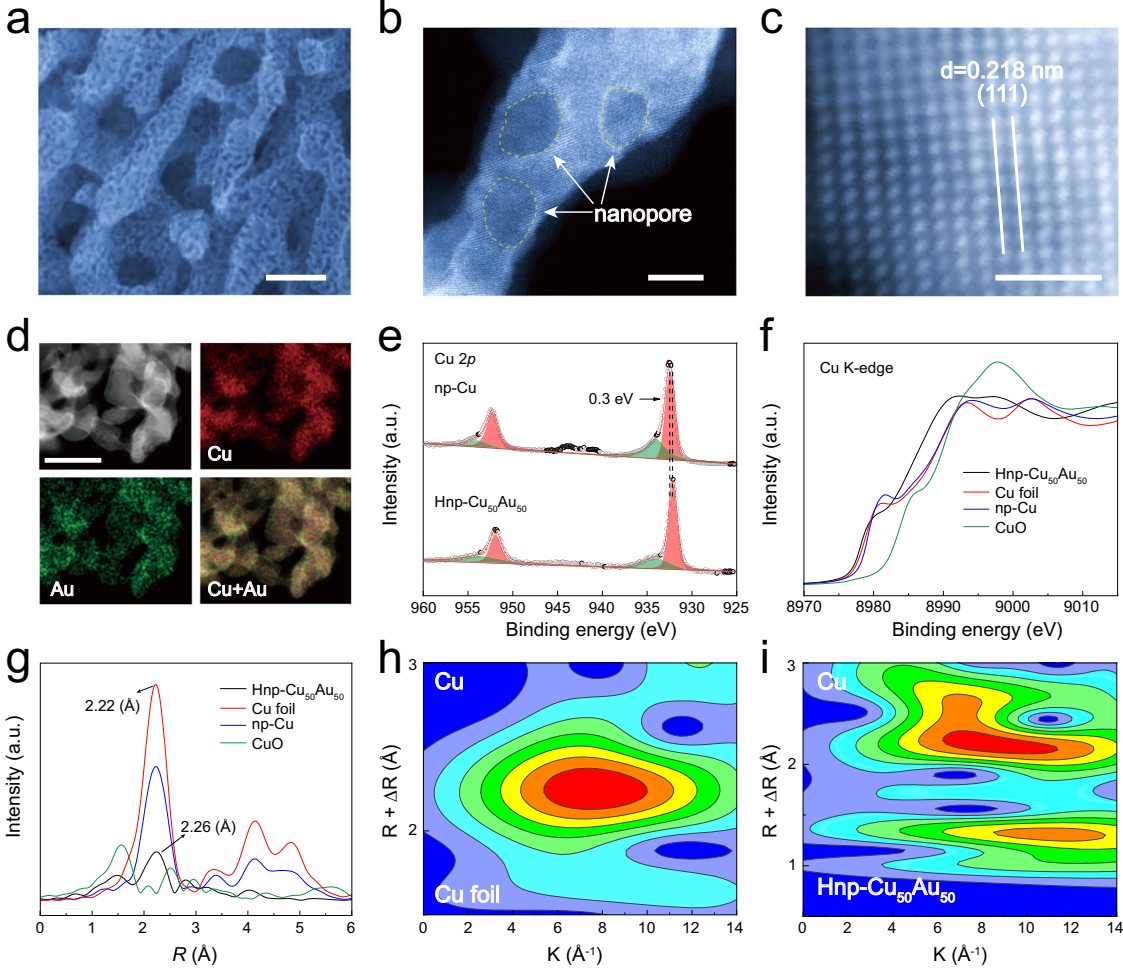

**Fig. 1 | Structural characterization of Hnp-Cu$_{50}$Au$_{50}$. a** SEM image of Hnp-Cu$_{50}$Au$_{50}$. **b** HAADF-STEM image of Hnp-Cu$_{50}$Au$_{50}$. **c** Atom-resolution HAADF image of Hnp-Cu$_{50}$Au$_{50}$. **d** STEM-EDX elemental mapping of Hnp-Cu$_{50}$Au$_{50}$. **e** XPS spectra of Hnp-Cu$_{50}$Au$_{50}$ and np-Cu in Cu 2$p$. **f** Normalized XANES at the Cu K-edge of Hnp-Cu$_{50}$Au$_{50}$, np-Cu, Cu foil, and CuO. **g** The corresponding FT-EXAFS spectra of the Cu K-edge derived from (**f**). WT of Cu K-edge EXAFS spectra of Cu foil (**h**) and Hnp-Cu$_{50}$Au$_{50}$ (**i**). Scale bars: **a** 100 nm, **b** 5 nm, **c** 1 nm, **d** 20 nm.

formation of abundant unsaturated coordination atoms and creation of defective surfaces[35]. Accordingly, the wavelet transform (WT) of Cu, Au EXAFS oscillation was further employed to analyze the coordination environment of Cu and Au. The bond lengths of Cu and Au in the WT contour plots of Hnp-Cu$_{50}$Au$_{50}$ are different from monometallic Cu (Fig. 1h, i) and Au due to alloying (Supplementary Fig. 7).

**Electrocatalytic performance for semi-hydrogenation of alkynes**

Electrochemical alkynes semi-hydrogenation performance of catalysts was evaluated in a divided three-electrode system by using 1 mmol of phenylacetylene as the model substrate with 1.0 M KOH and dioxane. Remarkably, the linear sweep voltammetry (LSV) curves of Hnp-Cu$_{50}$Au$_{50}$ display that the current density was increased slightly after adding phenylacetylene, implying the easier reduction of phenylacetylene in contrast to the competitive HER (Fig. 2a)[10], which further confirmation of the feasibility of electrochemical alkyne semi-hydrogenation[23]. After the electrolysis, the products at the cathode were analyzed by gas chromatography (GC) according to the standard calibration curves (Supplementary Fig. 8). The electrocatalytic semi-hydrogenation performance of hierarchically nanoporous CuAu alloys strongly depends on Cu/Au ratios (Supplementary Fig. 9). Notably, the Hnp-Cu$_{50}$Au$_{50}$ exhibits the best electrochemical alkyne semi-hydrogenation performance. Over 92% conversion (Con.) of phenylacetylene and 100% selectivity (Sel.) of styrene production were

observed from −0.3 to −0.6 V vs. RHE (Fig. 2b), indicating this potential-independent conversions and selectivity over Hnp-Cu$_{50}$Au$_{50}$. Impressively, exceeded 52% Faradaic efficiency (FE) of styrene could be achieved at the −0.4 V in 1 M KOH solution (Fig. 2b). Unfortunately, a more negative applied potential leads to a decreased FE for styrene from 52% at −0.4 V vs. RHE to 29% at −0.6 V vs. RHE. This trend can be attributed to the insufficient H* coverage at −0.3 V vs. RHE and more severe HER competition at highly negative potentials[36]. Therefore, we further enhance the H$_2$O dissociation ability by increasing the concentration of K$^+$ in the electrolyte to promote the semi-hydrogenation of alkynes (Supplementary Fig. 10). Accordingly, the high FE of 92% and the selectivity of 100% for styrene product were achieved at the potential of −0.4 V vs. RHE after introducing 1 M KCl into 1 M KOH electrolyte (Fig. 2c), which is much higher than the values for state-of-the-art electrocatalysts (Supplementary Table 1). We conducted a control experiment using nBu$_4$NCl instead of KCl added in 1 M KOH to investigate the effect of ionic strength for reaction performance. The experimental results indicate that the improvement in reaction performance is primarily due to the promotion of water dissociation by K$^+$ rather than the contribution of ion strength (Supplementary Fig. 11). Noticeably, both Hnp-Cu$_{50}$Au$_{50}$ and unimodal np-Cu$_{50}$Au$_{50}$ have high conversion of more than 90% at −0.4 V vs. RHE. However, Hnp-Cu$_{50}$Au$_{50}$ presents a higher selectivity and FE compared to np-Cu$_{50}$Au$_{50}$, indicating the enhanced activity aroused by induced a local

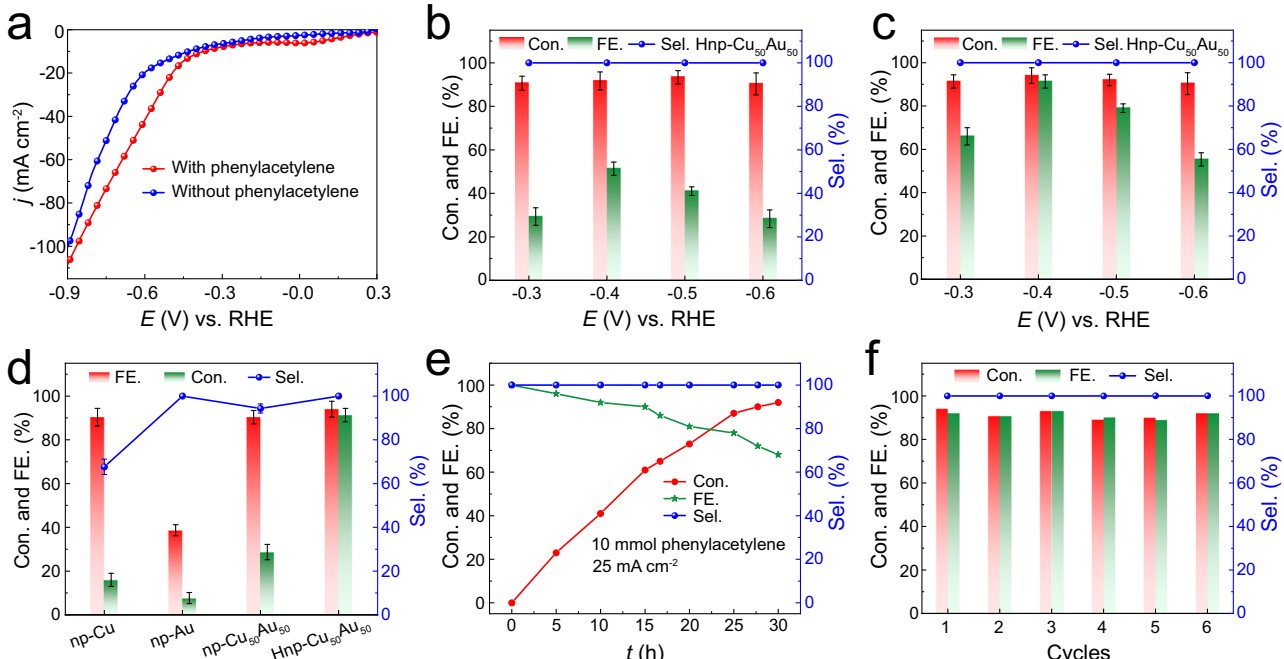

**Fig. 2 | Electrocatalytic performance of catalysts for alkyne semi-hydrogenation. a** LSV curves of Hnp-Cu$_{50}$Au$_{50}$ in 1 M KOH with and without 1 mmol phenylacetylene. **b** Potential-dependent conversions of phenylacetylene, selectivity and FE of styrene over Hnp-Cu$_{50}$Au$_{50}$ in 1 M KOH. **c** Potential-dependent conversions of phenylacetylene, selectivity and FE of styrene over Hnp-Cu$_{50}$Au$_{50}$ in 1 M KOH + 1 M KCl. **d** Con. of phenylacetylene, Sel., and FE of styrene over np-Cu, np-Au, np-Cu$_{50}$Au$_{50}$ and Hnp-Cu$_{50}$Au$_{50}$ catalysts at −0.4 V RHE in 1 M KOH + 1 M KCl. **e** The conversions of phenylacetylene, selectivity and FE of styrene over Hnp-Cu$_{50}$Au$_{50}$ in 1 M KOH + 1 M KCl with 10 mmol phenylacetylene in 25 mA cm$^{-2}$ current density. **f** Cyclic Conversion of phenylacetylene, selectivity and FE of styrene over Hnp-Cu$_{50}$Au$_{50}$ in 1 M KOH + 1 M KCl. The error bars represent the standard deviation for at least three independent measurements.

microenvironment with massive K$^+$ cations through the hierarchically nanoporous structure (Fig. 2d). In addition, unimodal np-Cu$_{50}$Au$_{50}$ keep much higher selectivity and FE than np-Cu, while it achieves a higher conversion and FE of alkynes than np-Au (Fig. 2d and Supplementary Fig. 12). In particular, the selectivity and FE of np-Cu catalyst gradually dropped with the negative bias potentials (Supplementary Fig. 12), ultimately reaching 42% selectivity and the FE of 11% at −0.6 V vs. RHE (Fig. 2d), which are attributed to the dominant HER and weak hydrogenation ability of Cu itself. Furthermore, when styrene was used as the substrate, unimodal np-Cu$_{50}$Au$_{50}$ exhibits lower conversion of styrene to over-hydrogenated alkane product phenylethane than np-Cu under identified conditions (Supplementary Fig. 13), confirming its intrinsic inhibition on the undesired phenylethane. These results indicate that Au alloyed with Cu can not only prevent the combination of H* to form H$_2$ and suppress the HER, but also inhibiting styrene over-hydrogenation. Moreover, the energy efficiency (EE) of the Hnp-Cu$_{50}$Au$_{50}$ for semi-hydrogenation of phenylacetylene is further evaluated, which shows a high EE of 42% at −0.4 V vs. RHE (Supplementary Fig. 14). More importantly, when subjecting a gram-scale (10 mmol) phenylacetylene into the reactor, Hnp-Cu$_{50}$Au$_{50}$ achieves excellent hydrogenation performance with 100% selectivity and 68% FE of styrene production at a current density of 25 mA cm$^{-2}$ (Fig. 2e). Furthermore, the high conversion yield and Faradaic efficiency were maintained after six cycles, suggesting the excellent durability of the Hnp-Cu$_{50}$Au$_{50}$ cathode under the electrochemical reduction conditions (Fig. 2f). The SEM, XRD, and XPS characterizations of the recycled Hnp-Cu$_{50}$Au$_{50}$ reveal no obvious morphological and structural variations, unambiguously confirming its robust stability (Supplementary Figs. 15–17).

To further assessment the universality of the electrochemical semi-hydrogenation approach for other key alkynes, the electrocatalytic performance of the Hnp-Cu$_{50}$Au$_{50}$ was measured in 1 M KOH (Dioxane/H$_2$O) containing different alkynes. As described in Supplementary Table 2, a series of aryl alkynes with electron-withdrawing or electron-donating substituents on the aryl ring can afford the corresponding alkenes with excellent selectivity under standard reaction conditions. The applicability and scalability of Hnp-Cu$_{50}$Au$_{50}$ confirm the promising potential in the application of electrocatalytic semi-hydrogenation of alkynes.

## Mechanistic studies on electrochemical semi-hydrogenation of alkyne over the Hnp-Cu$_{50}$Au$_{50}$

To better understand the potential mechanism of alloying in improving the electrocatalytic semi-hydrogenation performance of alkynes, operando XAS measurements using a homemade cell were performed to probe the electronic structure evolution of the Cu site on Hnp-Cu$_{50}$Au$_{50}$ during the electrocatalytic hydrogenation of phenylacetylene. During the operando XAS measurement, the potential was applied from open circuit voltage (OCV) to −0.5 V vs. RHE and then back to OCV. As the applied potential increases from OCV to −0.3 V and −0.5 V vs. RHE, the absorption edge of the Cu K-edge XANES spectra gradually shifts toward higher energy side, indicating an increase in the Cu valence state[33], there are abundant of reactants adsorbed on the Cu sites without subsequent desorption, which balances the reduction trend of cathodic voltage, the electron transfer from Cu to phenylacetylene and related intermediates during hydrogenation of phenylacetylene resulting in the further increase of oxidation state (Fig. 3a)[37–39]. In addition, compared to OCV, the Cu-Au bonds were significantly prolonged at −0.3 V and −0.5 V, further confirming the enhancing adsorption of phenylacetylene and intermediates on Cu, causing the distortion of Cu-Au bond configuration (Fig. 3b)[37,40]. Impressively, when the potential returns to OCV, the Cu valence state shift to the lower energies and the bond length of Cu-Au decreases, which can be persuasively ascribed to the rapid desorption process of reactants or reaction intermediates on the catalyst surface (Fig. 3a, b). In contrast with the Cu K-edge results

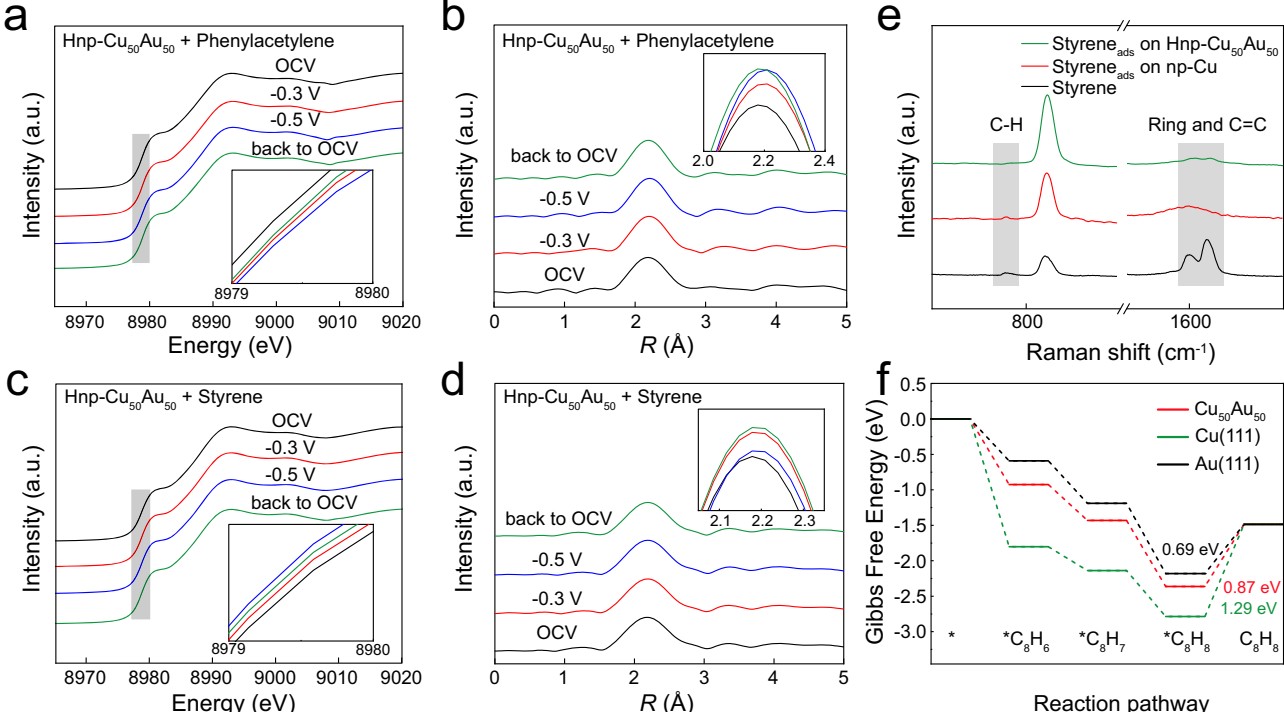

**Fig. 3 | Experimental and theoretical investigations on the reaction mechanism.** **a** Operando XANES spectra of Hnp-Cu$_{50}$Au$_{50}$ recorded at Cu K-edge (1 M KOH + phenylacetylene). **b** Cu K-edge FT-EXAFS spectra for Hnp-Cu$_{50}$Au$_{50}$ (1 M KOH + phenylacetylene). **c** Operando XANES spectra of Hnp-Cu$_{50}$Au$_{50}$ recorded at Cu K-edge (1 M KOH + styrene). **d** Cu K-edge FT-EXAFS spectra for Hnp-Cu$_{50}$Au$_{50}$ (1 M KOH + styrene). **e** In situ Raman tests for electrocatalytic hydrogenation of styrene over np-Cu and Hnp-Cu$_{50}$Au$_{50}$ at −0.4 V vs. RHE. **f** Gibbs free energy diagram for alkyne semi-hydrogenation reactions over Cu (111), Au (111), and Cu$_{50}$Au$_{50}$.

under phenylacetylene reaction conditions, the Cu K-edge XANES under styrene reaction conditions gradually shifts toward lower energy side as the potential from OCV to −0.3 V and −0.5 V vs. RHE, indicating the decrease in the Cu oxidation state and which is reduced to a certain extent (Fig. 3c). Meanwhile, the bond length of Cu-Au is no apparent changes (Fig. 3d), these could be due to the weak adsorption of styrene on the Hnp-Cu$_{50}$Au$_{50}$ catalyst. In situ Raman spectroscopy measurements were further carried out to validate the origin of the highly selective semi-hydrogenation of alkynes on Hnp-Cu$_{50}$Au$_{50}$ in a mixed solution of 1.0 M KOH/Dioxane. It is worth noting that the vibration peak at ~833 cm$^{-1}$ assigned to the C-H bond of dioxane. The C≡C stretching vibration peak of phenylacetylene is blue-shifted from 2108 to 2201 and 2218 cm$^{-1}$ and disappearing of the terminal C-H vibration peaks in the presence of np-Cu and Hnp-Cu$_{50}$Au$_{50}$ (Supplementary Fig. 18a), which implies the interaction between the alkynyl group and catalyst. Moreover, the vibrations of the C=C skeleton (1598 cm$^{-1}$) and C-H (1090 cm$^{-1}$ assigned to the in-plane rocking mode and 787 cm$^{-1}$ assigned to the out-of-plane bending) bonds of the benzene ring remain unchanged, indicating a negligible interaction with np-Cu and Hnp-Cu$_{50}$Au$_{50}$ (Supplementary Fig. 18b). Thus, phenylacetylene adsorbs on the np-Cu and Hnp-Cu$_{50}$Au$_{50}$ with only $\sigma$-alkynyl bonding adsorption mode[41]. Notably, the -CH=CH$_2$ vibration peak of styrene (1630 cm$^{-1}$) was disappeared, however, there is no shifts in the vibrations of C=C and C-H bonds of the benzene ring (1598 cm$^{-1}$ and 767 cm$^{-1}$) are observed on the surface of np-Cu, which suggests the interaction between the alkenyl group and np-Cu. In contrast, no clear shifts of these characteristic peaks with Hnp-Cu$_{50}$Au$_{50}$ (Fig. 3e). These results indicate that the alloying of Au with Cu can effectively weaken the adsorption of alkene on the catalyst surface to avoid over-hydrogenation into alkanes, thus boosting high alkene selectivity[20,41].

Density functional theory (DFT) calculations are further performed to rationalize the role of alloying in improving the

hydrogenation performance of phenylacetylene. Herein, Cu (111), Au (111), and Cu$_{50}$Au$_{50}$ alloy were used as DFT models. Figure 3f displays the Gibbs free energy profiles for the hydrogenation of phenylacetylene on Cu (111), Au (111), and Cu$_{50}$Au$_{50}$. The exothermal adsorptions of phenylacetylene imply the kinetically favorable phenylacetylene hydrogenation of these catalysts. Although the desorption of adsorbed phenylacetylene (*C$_8$H$_8$) from the surface of these catalysts to C$_8$H$_8$ are all exothermic, much less energy is required to drive the desorption of adsorbed phenylacetylene on Cu$_{50}$Au$_{50}$ than Cu (111), reflecting more favorable desorption of C$_8$H$_8$ on Cu$_{50}$Au$_{50}$. The projected density of states (PDOS) indicates that the d-band center of Cu$_{50}$Au$_{50}$ is more negative than that of Cu (111), indicating that alloying Cu with Au can lower the d-band center of Cu, thereby weakening the adsorption of C$_8$H$_8$ and making phenylacetylene more readily desorbed on Cu$_{50}$Au$_{50}$ (Supplementary Fig. 19). All of those means that Cu$_{50}$Au$_{50}$ alloy can weaken the adsorption of styrene, thereby avoiding the over-hydrogenation of styrene. In addition, the H* coupling barrier over Cu$_{50}$Au$_{50}$ is higher than the corresponding Cu, indicating that the introduction of Au into the Cu matrix can hinder the competitive HER (Supplementary Fig. 20). These calculated results reveal that Au alloying with Cu can not only suppress competitive HER, but also inhibit over-hydrogenation of C$_8$H$_8$, which is consistent with the previous experimental results.

In order to gain insight into the mechanism of the nanoporous structure on enhancing the electrocatalytic semi-hydrogenation of alkynes. We first compare the intrinsic activity of Hnp-Cu$_{50}$Au$_{50}$ and np-Cu$_{50}$Au$_{50}$ by ruling out the influence of different electrochemical surface areas (ECSA) (Supplementary Fig. 21). Obviously, the intrinsic activity of Hnp-Cu$_{50}$Au$_{50}$ catalyst for styrene production is higher than that of np-Cu$_{50}$Au$_{50}$ catalyst (Supplementary Fig. 22). Moreover, the electrochemical impedance (EIS) results show that Hnp-Cu$_{50}$Au$_{50}$ have a smaller charge transfer resistance ($R_{ct}$) than np-Cu$_{50}$Au$_{50}$,

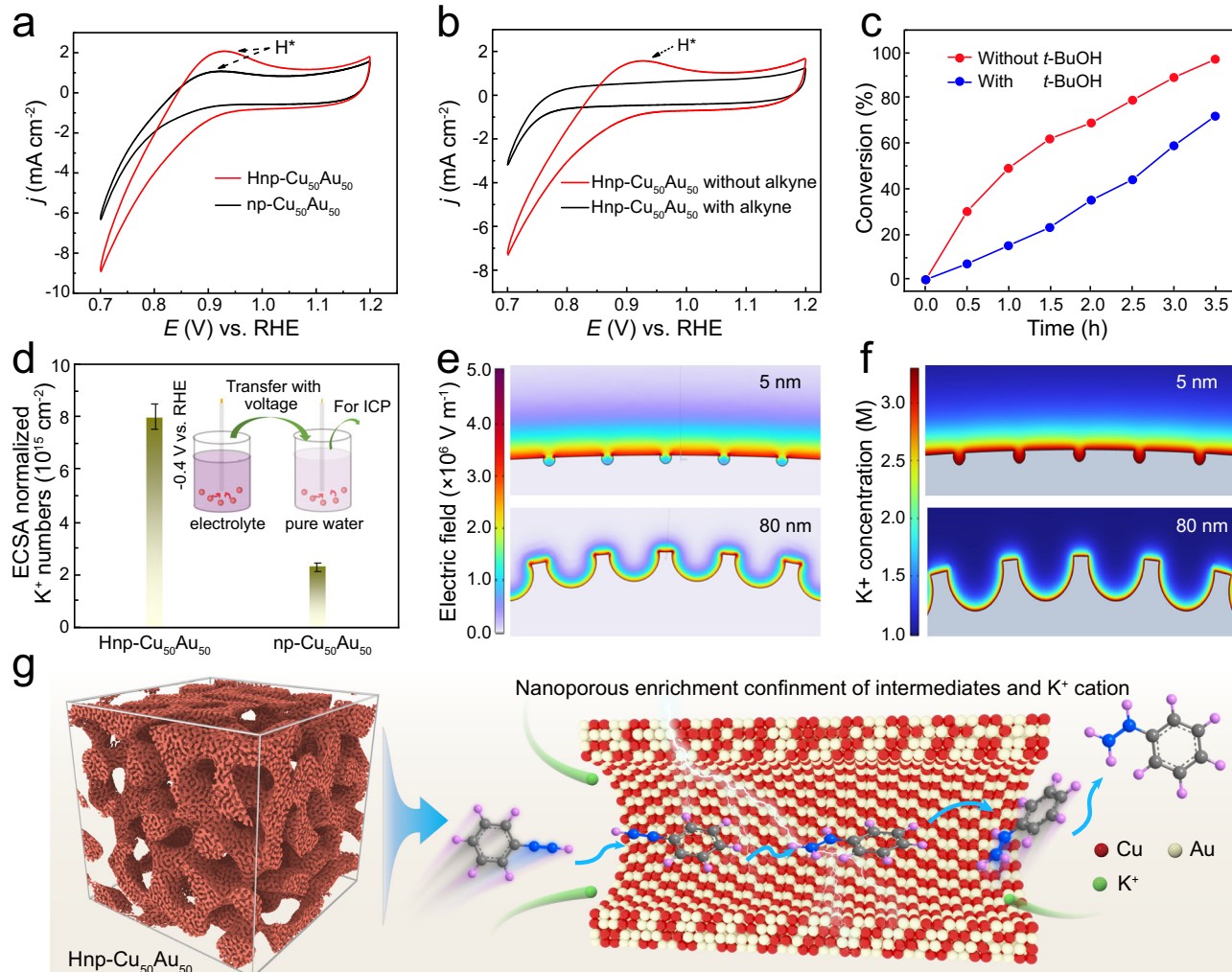

**Fig. 4 | The investigation of surface microenvironment of catalysts. a** CV curves of the np-Cu$_{50}$Au$_{50}$ and Hnp-Cu$_{50}$Au$_{50}$ in 1 M KOH with a scan rate of 100 mV s$^{-1}$. **b** CV curves of Hnp-Cu$_{50}$Au$_{50}$ with the addition of 1 mmol phenylacetylene, inset shows the enlarged H* desorption peak. **c** Time-dependent conversion change of phenylacetylene with and without *t*-BuOH over Hnp-Cu$_{50}$Au$_{50}$. **d** ECSA normalized K$^+$ number on np-Cu$_{50}$Au$_{50}$ and Hnp-Cu$_{50}$Au$_{50}$. The density of K$^+$ was measured by ICP-OES and the inset is the operation method. **e** Electric field on the surface of np-Cu$_{50}$Au$_{50}$ and Hnp-Cu$_{50}$Au$_{50}$ with different pore diameters of 5 and 80 nm. **f** Simulated concentrations and distributions of local K$^+$ around the surface of Hnp-Cu$_{50}$Au$_{50}$ and np-Cu$_{50}$Au$_{50}$ with various pore diameters of 5 and 80 nm. **g** The proposed reaction mechanism for alkyne semi-hydrogenation reaction over Hnp-Cu$_{50}$Au$_{50}$ cathode.

suggesting that the hierarchical structure promoted charge transfer kinetics (Supplementary Fig. 23). This result indicates that the electrocatalytic hydrogenation performance can be reasonably attributed to the presence of hierarchical structure with both macroporous and nanoporous networks. It has been widely accepted that the intensive local electric field induced by nanoporous structure could endow a distinct microenvironment near the catalyst's surface[24]. We conduct cyclic voltammograms (CV) tests in a 1 M KOH electrolyte over unimodal np-Cu$_{50}$Au$_{50}$ and Hnp-Cu$_{50}$Au$_{50}$ at 0.7 V to 1.2 V (vs. RHE) (Fig. 4a). Remarkably, the CV curve of Hnp-Cu$_{50}$Au$_{50}$ presents a significantly larger peak of surface-adsorbed H* compared to np-Cu$_{50}$Au$_{50}$, implying the existence of more H* on the Hnp-Cu$_{50}$Au$_{50}$[42]. After adding phenylacetylene, the H* desorption peaks became smaller and fully suppressed over Hnp-Cu$_{50}$Au$_{50}$ (Fig. 4b). We speculate that the decreased H* species is possibly due to consumption by phenylacetylene reduction or phenylacetylene occupation, thus resulting in decrease of the accessible sites for H* production[11]. Furthermore, when we added tertiary butanol (*t*-BuOH) to the reaction system, the conversion of phenylacetylene is significantly inhibited, again supporting an H* participated in hydrogenation reaction (Fig. 4c)[16]. Furthermore, the inductively coupled

plasma optical emission spectrometry (ICP-OES) was employed to quantify the adsorption of K$^+$ cations by the catalysts from the electrolyte solution. The result shows that the value of ECSA-normalized K$^+$ concentration on Hnp-Cu$_{50}$Au$_{50}$ is 3.6 times higher than that on unimodal np-Cu$_{50}$Au$_{50}$ (Fig. 4d), experimentally evidencing that the confinement effect from nanoporous structure could concentrate K$^+$ cations. The local confinement effect for the enhanced K$^+$ production on Cu$_{50}$Au$_{50}$ alloy was further elucidated by COMSOL Multiphysics finite-element-based simulations. Due to the nanoconfinement effect, the Cu$_{50}$Au$_{50}$ alloy with a surface pore diameter of 5 nm induced stronger local electric field compared with the pore diameter of 80 nm (Fig. 4e), thus resulting in higher local K$^+$ concentration near to nanopore sites (Fig. 4f). The concentrated K$^+$, which can produce more anion-hydrated cation networks [K$^+$(H$_2$O)$_n$], facilitating the dissociation of H$_2$O for forming H* (Supplementary Fig. 24), thus promoting the alkyne semi-hydrogenation[13,43]. Based on these results, the hierarchically nanoporous alloys with interconnected macroporous channels and abundant nanopore not only offer efficient mass transport and large active areas for the adsorption of phenylacetylene and water on the surface[44,45], but also enable enriched K$^+$ cation on the catalyst's surface via nanoconfinement

effect[46], which could ultimately facilitate the formation of more H* ($H_2O + e^- + * \rightarrow H* + OH^-$). Then, the H* reaction with the C≡C bond of a nearby phenylacetylene to form the carbon radical intermediates in the inner-sphere of the catalyst[47,48], and immediately couple with another H* to produce the styrene products. Subsequently, desorption of styrene on the alloy surface regenerates the catalytic sites for the next reaction cycle (Fig. 4g). The quasi-in situ electron paramagnetic resonance (EPR) measurements reveal the formation of the H radical and C radical during the electrocatalytic hydrogenation of phenylacetylene (Supplementary Fig. 25), further confirming that water can be reduced to form H* and undergo electro-hydrogenation reaction with adsorbed phenylacetylene.

## Discussion

In summary, the hierarchically nanoporous $Cu_{50}Au_{50}$ alloy was designed for highly efficient electrocatalytic semi-hydrogenation of alkynes to alkene. A high conversion (94%) of phenylacetylene, selectivity (100%) of styrene, and a recorded Faraday efficiency (92%) were achieved with $H_2O$ as the hydrogen source under environmental conditions via the combined effects of alloying and nanoconfinement engineering. Additionally, a gram-scale synthetic process with high FE of 68% is achieved at a current density of 25 mA cm$^{-2}$ for continuous styrene production. By virtue of operando XAS, in situ Raman measurements and DFT calculations, we confirmed that Au can effectively optimize the electronic structure of Cu, which can suppress the HER and weaken adsorption of alkenes to inhibit over-hydrogenation, thus resulting in a superior semi-hydrogenation selectivity to targeted alkenes. A series of controlled experiments and finite element simulation further revealed that the presence of nanoporous structure can induce a locally enhanced electric field to endow hydrated K$^+$ accumulation around the nanopores, accelerating the electrolysis of $H_2O$ to produce more H*, thereby promoting the electrocatalytic hydrogenation of alkynes. This work not only realizes the high yield and Faradaic efficiency for alkyne electrocatalytic semi-hydrogenation, but also provides a synergistic interaction of alloying and nanoconfinement strategy for other electrocatalytic reactions.

## Methods
### Materials
Aluminum foil (99.999%), Gold foil (99.99%), Copper foil (99.99%) were purchased from Beijing Jiaming Platinum Nonferrous Metals Co., Ltd. Phenylacetylene (97%), Styrene (99%), 1,4-dioxane (99.7%), Tert butanol (99.5%) were purchased from Adamas. KOH (95%) were purchased from Greagent. KCl (99.5%) and Nitric Acid (67%) were purchased from Sinopharm Co. Ltd. All chemicals were used without further purification.

### Preparation of CuAu nanoporous alloy.
$Al_{80}Cu_{18}Au_2$ ribbons were prepared via a melt-spinning process. The precursor ribbons were then chemically etched via two-step dealloying method. Firstly, 200 mg of $Al_{80}Cu_{18}Au_2$ ribbons was put into 200 ml of 2 M KOH aqueous solution to remove partial of Al in the ribbons. The etching reaction was kept at 30 °C for around 10 h until no apparent bubbles were observed. As shown in the X-ray powder diffraction (XRD) pattern, the obtained product from the first-step dealloying was composed of $Cu_{18}Au_2$, $Cu_2O$ and $Al_2Cu_3$ phases. The as-prepared product was washed with ultra-pure $H_2O$ for several times till the solution became neutral. After that, the second-step dealloying treatment was carried out by adding the obtained product into 200 ml of 0.5 M $HNO_3$ aqueous solution kept at 30 °C, $Cu_2O$ was thoroughly removed. At the same time, residues $Al_2Cu_3$ phase were selectively removed from the first

ordered pore structure. After 2 h, Hnp-$Cu_{50}Au_{50}$ alloy was obtained. After washing process, the product was dried in vacuum oven at 60 °C for 24 h for further structure characterization and electrocatalytic analysis. Hnp-$Cu_{70}Au_{30}$ and Hnp-$Cu_{35}Au_{65}$ alloys were prepared with the same reaction conditions except the reaction times of the second-step dealloying were changed to 1 and 3 h, respectively. For comparison, unimodal nanoporous Cu, Au, and $Cu_{50}Au_{50}$ were fabricated by dealloying $Al_{80}Cu_{20}$, $Al_{80}Au_{20}$, and $Al_{80}Cu_{10}Au_{10}$ ribbons in 2 M KOH at 30 °C for 10 h, respectively.

### Characterizations.
XRD patterns of the samples were conducted by using a Bruker D8 Advance X-ray diffractometer with Cu Kα radiation ($\lambda = 1.5418$ Å). Morphology and chemical composition were collected via MIR3 TESCAN SEM equipped with an Oxford energy dispersive X-ray spectroscope. HAADF-STEM and EDS mapping were conducted on a JEM-ARM 200F with double spherical aberration (Cs) correctors for both the probe forming and image-forming objective lenses at an accelerating voltage of 200 kV. The chemical state and composition of the samples were characterized using XPS (Thermo Scientific Escalab 250Xi) with an Al Kα monochromatic (150 W, 20 eV pass energy, 500 μm spot size). The content of K$^+$ were carried out via ICP-OES (Atom scan Advantage, Thermo Jarrell Ash).

### Operando X-ray absorption spectra.
The XAS experiments were carried out at Beamline 01C1 at Taiwan Synchrotron Radiation Research Center. A home-made Teflon electrochemical cell with electrochemical workstation (Ivium, Compact Stat.) was employed for operando XAS measurement under the sensitive fluorescence model. Adding phenylacetylene to a cell filled with 1.0 M KOH electrolyte. A graphite rod was used as the counter electrode and a Hg/HgO electrode was used as the reference electrode. The carbon paper loaded with catalyst as the working electrode was in contact with Kapton tape to the observation window of the cell. During the experimental measurement, different potentials of OCV, −0.3 and −0.5 V vs. RHE were applied to the system and each potential was maintained to collect spectra for 30 min. The acquired XAS data were processed using Athena software. All the voltage indicated in the "Methods" section has not been iR corrected.

### Electrochemical measurements.
Electrochemical measurements were carried out in a divided three-compartment electrochemical cell consisting of a working electrode, a carbon rod counter-electrode, and a Hg/HgO reference electrode. The cathode cell and anode cell containing 1.0 M KOH with or without 1 M KCl solution, respectively, were separated by a Nafion 117 proton exchange membrane. 1 mmol of alkynes dissolved in dioxane were added into the cathode and stirred to form a homogeneous solution (16 ml 1.0 M KOH + 1 M KCl and 4 ml dioxane). Then, chronoamperometry was carried out at a given constant potential and stirred until the starting substrates disappeared. The liquid products were extracted with ethyl acetate and then quantified by gas chromatography (Shimadzu, GC-2010 Plus) equipped with a flame ionization detector (FID). In a typical procedure of the fabrication of the working electrode, the catalyst ink was prepared by dispersing 5 mg of catalysts into a mixture solution of 0.48 ml ethanol and 20 μl of Nafion solution (5%, w/w, Alfa Aesar) with sonication for 30 min. Fifty μl of the electrocatalyst ink was loaded onto a carbon paper with an area of $1 \times 1$ cm$^2$ by drop-coating with the loading mass of catalyst is 0.50 mg cm$^{-2}$. The as-prepared catalyst film was dried at room temperature.

### Quantitative reductive product.
The conversion (%), selectivity (%), and Faradaic efficiency (FE, %) of alkenes were calculated using

Eqs. (1)−(3):

$$\text{Conversion}\,(\%) = \frac{\text{mol of formed alkene}}{\text{mol of initial alkyne}} \times 100\% \qquad (1)$$

$$\text{Selectivity}\,(\%) = \frac{\text{mol of formed alkene}}{\text{mol of consumed alkyne}} \times 100\% \qquad (2)$$

$$\text{FE}\,(\%) = \frac{nmF}{It} \times 100\% \qquad (3)$$

where $n$ = number of transferred electrons; $m$ = amount of substance; $F$ = Faraday's constant; $I$ = total current; $t$ = electrolysis time.

**Calculation of energy efficiency.** The energy efficiency (EE) was defined as the ratio of fuel energy to applied electrical power, which was calculated by:

$$EE_{styrene} = ((E^{\theta}_{OER} - E^{\theta}_{styrene}) \times FE_{styrene})/(E_{OER} - E_{styrene}) \qquad (4)$$

where $E^{\theta}_{styrene}$ represents the equilibrium potential of phenylacetylene electroreduction to styrene, which is calculated by DFT (0.49 V vs. RHE) (Supplementary Fig. 26)[6], $E^{\theta}_{OER}$ is the equilibrium potential of the oxygen evolution reaction (OER) (1.23 V vs. RHE), $FE_{styrene}$ is the Faradaic efficiency for styrene, and $E_{OER}$ and $E_{styrene}$ are the applied potentials.

**Surface-adsorbed K⁺.** Hnp-$Cu_{50}Au_{50}$ and np-$Cu_{50}Au_{50}$ were run in 1.0 M KOH at −0.4 V vs. RHE. After 2 min, the electrode was directly raised above the electrolyte and transferred into 5 ml pure water, during which the voltage was kept. After immersing in water, the voltage was removed to release any adsorbed K⁺ from the electrode. The transferred electrodes from the same aqueous solution without applying voltage were used as the blank background. Subsequently, the amount of K⁺ in the water was determined using an inductively coupled plasma optical emission spectrometer (ICP-OES, Atom scan Advantage, Thermo Jarrell Ash, USA). Finally, the amount of K⁺ in ultrapure water with the background deducted represents the true amount of K⁺ adsorbed on the surface of the Hnp-$Cu_{50}Au_{50}$ and np-$Cu_{50}Au_{50}$ catalysts. The obtained results were normalized by ECSA for comparison.

**Scavenge of high active H* with t-BuOH.** A total of 1 mmol of phenylacetylene was added into the electrolytic cell for the following semihydrogenation with/without t-BuOH. Chronoamperometry was carried out at a given constant potential −0.6 V vs. RHE for 210 min in 1 M KOH. The content of styrene was detected every 30 min.

**Calculation of the electrochemically active surface areas (ECSA).** ECSA was calculated from the equation as follow:

$$ECSA = \frac{C_{dl}}{C_s} \qquad (5)$$

The electrochemical double layer capacitance ($C_{dl}$) was measured by CV curves at different scan rates (Supplementary Fig. 21) and the general specific capacitance ($C_s$) found to be 60 μF cm⁻² in 1.0 M KOH.

**COMSOL Multiphysics simulations.** The electric field and K⁺ concentration within the vicinity of $Cu_{50}Au_{50}$ electrodes were simulated by solving the Poisson-Nernst-Planck equations using the COMSOL Multiphysics finite-element-based solver (https://www.comsol.com/). The structure models for catalyst particles with representative pore sizes (5.0 nm and 80 nm) were constructed based on the

experimental SEM images to perform finite-element-method (FEM) simulations.

The electric field ($E$) distribution was described by the following equation:

$$E = -\nabla V \qquad (6)$$

$$\rho = \varepsilon_r \varepsilon_0 \nabla \cdot E \qquad (7)$$

where $V$, $\rho$, $\varepsilon_r$, and $\varepsilon_O$ represent the applied potential bias, charge density, dielectric in vacuum and materials, respectively. $E$ was the negative gradient of the electric potential.

We choose the bulk solution as the grounding condition for the electrolyte potential:

$$\Phi = 0 \qquad (8)$$

The ion absorption behavior was described by the Nernst−Planck equations:

$$\nabla \cdot \left( Di\nabla ci + \frac{Di z i F}{RT} ci \nabla \psi \right) = 0 \qquad (9)$$

where $c$, $D$, $z$ are the ion concentrations, diffusion coefficients, ion valences, respectively. In addition, $F$, $R$, and $T$ represent the Faraday's constant, gas constant, and absolute temperature ($T$ = 293.15 K), respectively, and $\psi$ is the electrostatic potential that satisfies the Poisson equation[49].

**Computational methods.** First-principles calculations were implemented using Vienna Ab-initio Simulation Package (VASP 5.4.4)[50,51], with the Perdew-Burke-Ernzerhof (PBE) exchange-correlation functional of generalized gradient approximation (GGA). The basis set utilized projector-augmented-wave pseudopotential (PAW) method, and the energy cut off was set at 400 eV[52,53]. Convergence was assumed when forces on each atom was less than 0.02 eV/Å and the self-consistent field (SCF) tolerance was 10⁻⁵ eV in the geometry optimization. The DFT-D3 method with Grimme's scheme was employed to correct the van der Waals interactions[54]. In the calculations, the pure Cu and Au surface were modeled by three-layer (111) fcc slabs with a 6 × 6 supercell, and a 4 × 4 supercell of CuAu (110) with three atomic layers was considered for the CuAu Alloy. For the Brillouin zone integration, a Monkhorst-Pack k-point mesh of 2 × 2 × 1 was employed. Here, the top two atomic layers were relaxed, and all the atoms at the bottom were frozen. To avoid the interactions between periodic structure, the vacuum space of 20 Å was employed along the z direction.

As an indicator for each elemental step of phenylacetylene semihydrogenation, the Gibbs free energy ($G$) is calculated by:

$$G = E + E_{ZPE} + \int C_p dT - TS \qquad (10)$$

where $E$ is the DFT-optimized total energy, $E_{ZPE}$ is the zero-point vibrational energy, $C_p$ is the heat capacity, $T$ = 298.15 K is the temperature and $S$ is the entropy. All corrections of Gibbs free energy were obtained using VASPKIT (v.1.1.2) software[55].

## Data availability

The data supporting the findings of this study are available from the corresponding authors upon reasonable request. The source data underlying Figs. 1−4 are provided as a Source Data file. Source data are provided with this paper.

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

## Acknowledgements

This work was supported by the National Natural Science Foundation of China (nos. 52371221 and U23A20554 to Y.W.T.), the Yunnan Provincial Technology Innovation Talent Development Program (no. 202105AD160028 to X.Z.), and Independent Research Project of State Key Laboratory of Advanced Design and Manufacturing Technology for Vehicle Body (no. 72365004 to Y.W.T.). The Raman and STEM tests were performed at Analytical Instrumentation Center of Hunan University.

## Author contributions

Y.W.T. conceived and directed the project. L.H.M. carried out key experiments. Z.W. performed theoretical calculations. C.W.K. and Y.R.L. contributed to the XAS measurements and analyses of the XAS experiment results. J.M. and P.F.H. performed COMSOL Multiphysics simulations. N.Z., X.Z. and M.P. contributed to the data analysis. Y.W.T. and L.H.M. wrote the manuscript with input from all other authors. All authors discussed the results and commented on the manuscript.

## Competing interests

The authors declare no competing interests.
