## [Peer Review File · Nature Communications]

Alloying and Confinement effects on Hierarchically Nanoporous CuAu for Efficient Electrocatalytic Semi-Hydrogenation of Terminal AlkynesREVIEWER COMMENTS

Reviewer #1 (Remarks to the Author):

In this work, the authors reported a hierarchically nanoporous Cu₅₀Au₅₀ (Hnp-Cu₅₀Au₅₀) alloy for semi-hydrogenation of alkynes with high yield and a record Faradaic efficiency via the combining of alloying and confinement effect. The authors utilized comprehensive and impressive in-situ tools and finite element simulations to explore the intrinsic reasons of the outstanding performances, achieving a very convincing conclusion. The results are solid, and the conclusion can potentially extend to other similar systems for guiding future electrocatalysis design. I have some issues require authors to address and suggest revision for publication of this manuscript in Nature Communications.

Major questions:

1. The FT-EXAFS spectrum for Hnp-Cu₅₀Au₅₀ shows the much lower Cu-Au peak intensity in comparison with Cu foil (Fig 1g), I think the main reason is due to the grains become smaller.
2. The Hnp-Cu₅₀Au₅₀ exhibits a prominent peak at ~1.5 Å from the Cu-O bonds (Fig 1g). However, it disappeared during OCV testing (Fig 3b, Fig 3d). Please provide additional explanation.
3. The authors should check the grammar and writing standards in the manuscript.
4. The absorption edge positions of Cu are mislabeled in operando XANES (Fig 3a, Fig 3c) (see 10.1038/s41467-018-02819-7).
5. The authors need provide the full spectrum of XPS.
6. There should have a carbon peak in the XRD test results of the Hnp-Cu₅₀Au₅₀ after the reaction (Fig S13).

Reviewer #2 (Remarks to the Author):

This communication by Y. Tan and colleagues describe the synergistic effect of hierarchical nanopores and Cu-Au alloy for the electrocatalytic semi-hydrogenation of terminal alkynes. The nanoconfinement strategy achieves high efficiency (including high FE, which is sometimes a neglected parameter in this field) thanks to a higher local K⁺ concentration. The study is very well presented, nicely combining insights from various spectroscopic (including operando XAS) and electrochemical techniques as well as theoretical (DFT) results to support the catalytic performance, making it an important read for the electrocatalytic hydrogenation community.

Therefore, I would recommend the publication after the following points have been addressed:

- 1) Regarding the scope of the reaction, the authors should clarify that it applies only to terminal alkynes, unless proven otherwise. In my opinion, this should even be stated in the title of the manuscript.
- 2) In the introduction, the authors contrast hydrogenation based on grey hydrogen (methane reforming) with ECH. I think it's important to recall that hydrogen can be produced by electrolysis of water, as this is a mature process, unlike ECH. In my opinion, the advantage of the latter lies in the economy of the process steps and its energy-saving potential, rather than in defossilization.
- 3) l. 48: "deleterious organic solvent". I see no reason why electrocatalytic hydrogenation (ECH)

should preclude the use of organic solvents. In fact, they are often needed to solubilize organic substrates, and dioxane is used in this work which, in my opinion, is a “deleterious organic solvent” (see also point 11).

4) The reference electrode used in this study (Hg/HgO) is not standard and the potential values are therefore more difficult for the reader to grasp. I would recommend converting the potential to ferrocenium/ferrocene standard or silver-based reference electrodes, or at least giving the conversion constant relative to these couples/electrodes.

5) I find the positive effect of a higher K⁺ concentration on the system remarkable. However, I would suggest probing the contribution of ionic strength. For instance, another salt that does not contain K⁺, such as +1M nBu₄NCl + 1M KOH electrolyte, can be added (instead of KCl).

6) l. 232: “return to their initial states” I think the authors should clarify this statement, as the green and black curves do not overlap in either case (insets in figures 3a and 3c).

For Raman analyses (Figure 3e), it is surprising that styrene adsorbed on np-Cu, and in particular the signals associated with the C=C bond, are not shifted relative to styrene alone, contrary to what is observed for alkynes [J. Am. Chem. Soc. 2022, 144, 19456]. Can the authors comment on this point?

7) It would also be interesting to perform the same Raman analyses in the presence of the substrate (i.e. phenylacetylene) to obtain further information on the binding of this species to the catalyst.

8) As constant-potential electrolysis has been carried out, it would be very interesting if the authors could estimate the overpotential applied in this work (at -1.3V, for instance). It is particularly important for estimating the energy efficiency of a system (beyond FE).

9) It would also be very useful, in my opinion, to have the over-reduction pathways computed by DFT (to complete figure 3f) to see if they can also support the high selectivity observed.

10) l.310 to 313: When describing the mechanism, it is not always clear whether the proposed steps take place in the inner or outer-sphere of the catalyst, particularly for radical intermediates, although it is probably difficult to distinguish for each of the steps.

11) l. 324: “environmental conditions” I don't find this statement very clear. Does it mean environmentally friendly? If so, this is far from being the case when using highly basic aqueous media combined with relatively toxic dioxane.

Also a few minor points:

12) The three terms Faraday, Faradic and Faradaic efficiencies are used throughout this manuscript and in the ESI. I suggest using only the last one.

13) l. 329: “alkynes” should be read as “alkenes”

Responses to the Referees' Comments

We thank the referees for their valuable comments and positive endorsement to our manuscript. We have carefully considered the referees' comments and revised the manuscript accordingly. Our responses and corresponding revisions are as follows:

Reviewer #1

In this work, the authors reported a hierarchically nanoporous Cu₅₀Au₅₀ (Hnp-Cu₅₀Au₅₀) alloy for semi-hydrogenation of alkynes with high yield and a record Faradaic efficiency via the combining of alloying and confinement effect. The authors utilized comprehensive and impressive in-situ tools and finite element simulations to explore the intrinsic reasons of the outstanding performances, achieving a very convincing conclusion. The results are solid, and the conclusion can potentially extend to other similar systems for guiding future electrocatalysis design. I have some issues require authors to address and suggest revision for publication of this manuscript in Nature Communications.

Response: Thank you for your positive comments on our manuscript. We have revised our manuscript accordingly.

Comment 1. The FT-EXAFS spectrum for Hnp-Cu₅₀Au₅₀ shows the much lower Cu-Au peak intensity in comparison with Cu foil (Fig 1g), I think the main reason is due to the grains become smaller.

Response: Thank you for your comments. Following the comment, we have carefully revised our manuscript and the details are listed below:

Importantly, the FT-EXAFS spectrum for Hnp-Cu₅₀Au₅₀ shows the much lower Cu-Au peak intensity in comparison with Cu foil, resulting from the grains become smaller, which could formation of abundant unsaturated coordination atoms and creation of defective surfaces.

Comment 2. The Hnp-Cu₅₀Au₅₀ exhibits a prominent peak at ~1.5 Å from the Cu-O

bonds (Fig 1g). However, it disappeared during OCV testing (Fig 3b, Fig 3d). Please provide additional explanation.

Response: We appreciate the questions from this reviewer. This is attributed to the surface oxide, which is removed by CV scanning before conducting the OCV testing.

Comment 3. The authors should check the grammar and writing standards in the manuscript.

Response: Many thanks for these comments. We have revised our manuscript accordingly.

Comment 4. The absorption edge positions of Cu are mislabeled in operando XANES (Fig 3a, Fig 3c) (see 10.1038/s41467-018-02819-7).

Response: We appreciate you for this insightful and constructive recommendation. Following your comments, we adjusted the labeled absorption edge positions of Cu and added the magnified rising edge XANES regions in the revised manuscript for the better comparison (Figure 3a, 3c).

Figure R1. (a) Operando XANES spectra of Hnp-Cu₅₀Au₅₀ recorded at Cu K-edge (1 M KOH + phenylacetylene). (c) Operando XANES spectra of Hnp-Cu₅₀Au₅₀ recorded at Cu K-edge (1 M KOH + styrene).

Comment 5. The authors need provide the full spectrum of XPS.

Response: We appreciate the reviewer for this nice suggestion. Following this

comment, we have supplement the XPS full spectrum data as follows (see Figure S6 in revised supporting information).

Figure R2. XPS full spectra of Hnp-Cu₅₀Au₅₀.

Comment 6. There should have a carbon peak in the XRD test results of the Hnp-Cu₅₀Au₅₀ after the reaction.

Response: Thank you for the kind suggestion. Following this comment, we have supplement the XRD patterns of Hnp-Cu₅₀Au₅₀ before and after reaction from 10 to 80 degrees (see Figure S15 in revised supporting information).

Figure R3. XRD patterns of Hnp-Cu₅₀Au₅₀ before and after reaction.

Reviewer #2

This communication by Y. Tan and colleagues describe the synergistic effect of hierarchical nanopores and Cu-Au alloy for the electrocatalytic semi-hydrogenation of terminal alkynes. The nanoconfinement strategy achieves high efficiency (including high FE, which is sometimes a neglected parameter in this field) thanks to a higher local K⁺ concentration. The study is very well presented, nicely combining insights from various spectroscopic (including operando XAS) and electrochemical techniques as well as theoretical (DFT) results to support the catalytic performance, making it an important read for the electrocatalytic hydrogenation community. Therefore, I would recommend the publication after the following points have been addressed:

Response: Thank you for your positive comments on our manuscript. We have revised our manuscript accordingly.

Comment 1. Regarding the scope of the reaction, the authors should clarify that it applies only to terminal alkynes, unless proven otherwise. In my opinion, this should even be stated in the title of the manuscript.

Response: We sincerely appreciate your valuable comment. Following your comments, we have revised the title of the paper as follows:

“Coupled Alloying and Confinement effects on Hierarchically Nanoporous CuAu for Efficient Electrocatalytic Semi-Hydrogenation of Terminal Alkynes”

Comment 2. In the introduction, the authors contrast hydrogenation based on grey hydrogen (methane reforming) with ECH. I think it's important to recall that hydrogen can be produced by electrolysis of water, as this is a mature process, unlike ECH. In my opinion, the advantage of the latter lies in the economy of the process steps and its energy-saving potential, rather than in defossilization.

Response: We appreciate the suggestions from this reviewer. We sincerely appreciate the reviewer for this reminding. We have revised the manuscript as follows:

“However, a large part of the gaseous H₂ source of the TCH processes is produced by the fossil fuel-based steam reforming process that inevitably results in high energy

consumption and releases massive amounts of CO₂⁷. Alternatively, electrochemical hydrogenation technology, powered by renewable and clean energy, which would be more attractive and sustainable due to its low cost, high safety and environment friendly⁸.”

Ref. [7] Han, G., Li, G. & Sun, Y. Electrocatalytic dual hydrogenation of organic substrates with a Faradaic efficiency approaching 200%. *Nat. Catal.* **6**, 224-233 (2023).

Ref. [8] Tang, C., Zheng, Y., Jaroniec, M. & Qiao, S. Z. Electrocatalytic refinery for sustainable production of fuels and chemicals. *Angew. Chem. Int. Ed.* **60**, 19572-19590 (2021).

Comment 3. 48: “deleterious organic solvent”. I see no reason why electrocatalytic hydrogenation (ECH) should preclude the use of organic solvents. In fact, they are often needed to solubilize organic substrates, and dioxane is used in this work which, in my opinion, is a “deleterious organic solvent” (see also point 11).

Response: Thank you very much for this comment. We are sorry for the loose of this description. What we want to express is deleterious organic hydrogen sources (e.g., formic acid) causes several concerns regarding the safety and sustainability. And we have revised our manuscript as follows:

“In this regard, a large amount of surface active hydrogen (H*) generated by water electrolysis can directly utilized as a sustainable hydrogen source to electrocatalytic transfer hydrogenation, avoiding the use of hazardous hydrogen or deleterious organic hydrogen sources (e.g., formic acid)⁹.”

Ref. [9] Liu, C., Wu, Y., Zhao, B. & Zhang, B. Designed nanomaterials for electrocatalytic organic hydrogenation using water as the hydrogen source. *Accounts. Chem. Res.* **56**, 1872-1883 (2023).

Comment 4. The reference electrode used in this study (Hg/HgO) is not standard and the potential values are therefore more difficult for the reader to grasp. I would recommend converting the potential to ferrocenium/ferrocene standard or silver-based reference electrodes, or at least giving the conversion constant relative to these

couples/electrodes.

Response: We appreciate the reviewer for this nice suggestion. We have converted the potential to reversible hydrogen electrode (RHE) in manuscript and ESI by the following Nernst equation:

$$E(\text{vs RHE}) = E(\text{vs Hg/HgO}) + 0.098 \text{ V} + 0.0591 \times \text{pH}$$

1.0 M KOH (pH = 13.6)

Comment 5. I find the positive effect of a higher K⁺ concentration on the system remarkable. However, I would suggest probing the contribution of ionic strength. For instance, another salt that does not contain K⁺, such as +1M nBu₄NCl + 1M KOH electrolyte, can be added (instead of KCl).

Response: We appreciate you for this constructive recommendation. Following your insightful comments, we conducted a control experiment using nBu₄NCl instead of KCl added in 1 M KOH to investigate the effect of ionic strength for reaction performance. The experimental results indicate that the improvement in reaction performance is primarily due to the promotion of water dissociation by K⁺ rather than the contribution of ion strength (Figure R4). The results are added in Figure S11.

Figure R4. Conversions of phenylacetylene, selectivity and FEs of styrene over Hnp-Cu₅₀Au₅₀ in 1 M KOH + 1 M KCl and 1 M KOH + 1 M nBu₄NCl.

Comment 6. 232: “return to their initial states” I think the authors should clarify this statement, as the green and black curves do not overlap in either case (insets in figures 3a and 3c).

For Raman analyses (Figure 3e), it is surprising that styrene adsorbed on np-Cu, and in particular the signals associated with the C=C bond, are not shifted relative to styrene alone, contrary to what is observed for alkynes [J. Am. Chem. Soc. 2022, 144, 19456]. Can the authors comment on this point?

Response: We sincerely thank you for these comments. For your convenience, we provide a point-by-point response to these comments:

(1) We are sorry for the unclear description. Following this comment, we revised this description as following:

“Impressively, when the potential returns to OCV, the Cu valence state shift to the lower energies and the bond length of Cu-Au decreases, which can be persuasively ascribed to the rapid desorption process of reactants or reaction intermediates on the catalyst surface.”

(2) We are sorry for the wrong description in this part. Following these comments, we retested the in situ Raman (Figure R5) and revised this part as following:

“Notably, the -CH=CH₂ vibration peak of styrene (1630 cm⁻¹) was vanished, however, there is no shifts in the vibrations of C=C and C-H bonds of the benzene ring (1598 cm⁻¹ and 767 cm⁻¹) are observed on the surface of np-Cu, which suggests the interaction between the alkenyl group and np-Cu. In contrast, no clear shifts of these characteristic peaks with Hnp-Cu₅₀Au₅₀ (Figure 3e). These results indicate that the alloying of Au with Cu can effectively weaken the adsorption of alkene on the catalyst surface to avoid over-hydrogenation into alkanes, thus boosting high alkene selectivity^{20,41}.”

Figure R5. In situ Raman tests for electrocatalytic hydrogenation of styrene over np-Cu and Hnp-Cu₅₀Au₅₀ at -0.4 V vs. RHE.

Ref. [20] Bai, L. et al. Efficient industrial-current-density acetylene to polymer-grade ethylene via hydrogen-localization transfer over fluorine-modified copper. *Nat. Commun.* **14**, 8384 (2023).

Ref. [41] Li, H. et al. σ -alkynyl adsorption enables electrocatalytic semihydrogenation of terminal alkynes with easy-reducible/passivated groups over amorphous PdS_x nanocapsules. *J. Am. Chem. Soc.* **144**, 19456-19465 (2022).

Comment 7. It would also be interesting to perform the same Raman analyses in the presence of the substrate (i.e. phenylacetylene) to obtain further information on the binding of this species to the catalyst.

Response: We sincerely thank you for this nice suggestion. Following the comment, we performed the in situ Raman analyses in the presence of the phenylacetylene (Figure R6) and supplement the information in our manuscript (Figure S18) as below:

“The C \equiv C stretching vibration peak of phenylacetylene is blue-shifted from 2108 to 2201 and 2218 cm⁻¹ and vanishing of the terminal C-H vibration peaks in the presence of np-Cu and Hnp-Cu₅₀Au₅₀, which suggests the interaction between the alkynyl group and catalysts. Additionally, the vibrations of the C=C skeleton (1598 cm⁻¹) and C-H (1090 cm⁻¹ for the in-plane rocking mode and 787 cm⁻¹ for the out-of-plane bending)

bonds of the benzene ring remain unchanged, indicating a negligible interaction with np-Cu and Hnp-Cu₅₀Au₅₀. Thus, phenylacetylene adsorbs on the np-Cu and Hnp-Cu₅₀Au₅₀ with only σ -alkynyl bonding adsorption mode⁴¹. "

Figure R6. In situ Raman tests in a mixed 1.0 M KOH/Diox solution for electrocatalytic hydrogenation of phenylacetylene over np-Cu and Hnp-Cu₅₀Au₅₀ at -0.4 V vs. RHE.

Ref. [41] Li, H. et al. σ -Alkynyl Adsorption Enables Electrocatalytic Semihydrogenation of Terminal Alkynes with Easy-Reducible/Passivated Groups over Amorphous PdSx Nanocapsules. *J. Am. Chem. Soc.* **144**, 19456–19465 (2022).

Comment 8. As constant-potential electrolysis has been carried out, it would be very interesting if the authors could estimate the overpotential applied in this work (at -1.3 V, for instance). It is particularly important for estimating the energy efficiency of a system (beyond FE).

Response: Thank you for the kind suggestion.

The energy efficiency was defined as the ratio of fuel energy to applied electrical power, which was calculated by:

$$EE_{styrene} = ((E_{OER}^{\theta} - E_{styrene}^{\theta}) \times FE_{styrene}) / (E_{OER} - E_{styrene})$$

where $E_{styrene}^{\theta}$ represents the equilibrium potential of phenylacetylene electroreduction to styrene, which is calculated by DFT (0.49 V vs. RHE) (Figure R7 and Figure S26)^[6], E_{OER}^{θ} is the equilibrium potential of the oxygen evolution reaction (OER) (1.23 V vs. RHE), $FE_{styrene}$ is the Faradaic efficiency for styrene, and E_{OER} and $E_{styrene}$ are the

applied potentials.

According to the above formula, the energy efficiency is calculated, and we have added the following sentences to our manuscript:

Moreover, the energy efficiency (EE) of the Hnp-Cu₅₀Au₅₀ for semi-hydrogenation of phenylacetylene is further evaluated, which shows a high EE of 42% at -0.4 V vs. RHE (Figure R8 and Figure S14).

Figure R7. (a) Standard Gibbs free energies of H₂(g), C₈H₆, C₈H₈ and H₂O(l). (b) The standard equilibrium potential of phenylacetylene-to-styrene conversion routes.

Figure R8. The energy efficiency of Hnp-Cu₅₀Au₅₀ for semi-hydrogenation of phenylacetylene at different applied potentials.

Ref. [6] Shi, R. et al. Room-temperature electrochemical acetylene reduction to ethylene with high conversion and selectivity. *Nat. Catal.* **4**, 565–574 (2021).

Comment 9. It would also be very useful, in my opinion, to have the over-reduction pathways computed by DFT (to complete figure 3f) to see if they can also support the

high selectivity observed.

Response: We are grateful to the reviewer for this nice comment. We have completed the over-reduction pathway of phenylacetylene hydrogenation through DFT. For the pathway from styrene to phenylethane ($*C_8H_8 \rightarrow *C_8H_9 \rightarrow *C_8H_{10}$), we found that the barrier for hydrogenation of styrene is smaller than the energy of styrene required for desorption into free molecules. We speculate that this is because our actual catalyst is an alloy phase metal of Cu and Au, and the atomic occupancy of its FCC structure has a certain degree of disorder. It is difficult to construct a matching model structure through DFT calculations, and the computational workload is significant. We will perform more comprehensive and detailed calculations in the follow-up work. Many thanks for your understanding. However, our current results (Figure R9 and Figure S19), are sufficient to explain the excellent hydrogenation performance of $Cu_{50}Au_{50}$ from phenylacetylene to styrene. The Cu (111) surface exhibits strong adsorption of C_8H_x species, making it easier to hydrogenate to phenylethane. However, the addition of Au atoms weakens the adsorption effect on the Cu surface, which can regulate a moderate surface of hydrogenation to the alkene. Meanwhile, the provision of H protons on the Au (111) surface is more difficult, while the introduction of Cu can provide a moderate hydrogen source, which can also prevent over hydrogenation to phenylethane.

Figure R9. Gibbs free energy diagram for alkyne semi-hydrogenation reactions over Cu (111), Au (111), and $Cu_{50}Au_{50}$.

Comment 10. 310 to 313: When describing the mechanism, it is not always clear whether the proposed steps take place in the inner or outer-sphere of the catalyst, particularly for radical intermediates, although it is probably difficult to distinguish for each of the steps.

Response: We thank the reviewer for presenting this nice question.

For outer-sphere reactions, the substrates, intermediates, and products do not interact strongly with the electrode material and electron transfer occurs by tunnelling across a solvation layer, while in an inner-sphere reaction there is a strong interaction between the substrate and the electrode surface^[47,48]. According to the results of in situ Raman and operando XAS, there is a strong interaction between phenylacetylene and the catalyst. Therefore, the reaction steps take place in the inner-sphere of the catalyst. Following this comment, we revised our manuscript as follows:

Then, the H* reaction with the C≡C bond of a nearby phenylacetylene to form the carbon radical intermediates in the inner-sphere of the catalyst, and immediately couple with another H* to produce the styrene products. Subsequently, desorption of styrene on the alloy surface regenerates the catalytic sites for the next reaction cycle

Ref. [47] Zhou, H. et al. Electrocatalytic oxidative upgrading of biomass platform chemicals: from the aspect of reaction mechanism. *Chem. Commun.* **58**, 897 (2022).

Ref. [48] Zhang, P. & Sun, L. Electrocatalytic hydrogenation and oxidation in aqueous conditions. *Chin. J. Chem.* **38**, 996—1004 (2020).

Comment 11. 324: “environmental conditions” I don't find this statement very clear. Does it mean environmentally friendly? If so, this is far from being the case when using highly basic aqueous media combined with relatively toxic dioxane.

Response: We appreciate you for this valuable recommendation. We are sorry for the loose of this description. Compared with the high temperature and pressure conditions of thermochemical hydrogenation, electrochemical hydrogenation was carried out at room temperature and atmospheric pressure. What we want to express is that the reaction conditions are mild.

Comment 12. The three terms Faraday, Faradic and Faradaic efficiencies are used throughout this manuscript and in the ESI. I suggest using only the last one.

Response: We appreciate you for this valuable recommendation. We have made revisions in the manuscript and ESI.

Comment 13. 329: “alkynes” should be read as “alkenes”

Response: We really appreciate your reminder. Following your comments, we revised it as following:

By virtue of operando XAS, in situ Raman measurements, and DFT calculations, we confirmed that the incorporation of Au atoms into the Cu matrix could effectively optimize the electronic structure of alloys, which can suppress the HER and weaken adsorption of alkenes to inhibit over-hydrogenation, thus resulting in a superior semi-hydrogenation selectivity to targeted alkenes.

REVIEWER COMMENTS

Reviewer #1 (Remarks to the Author):

My comments have been well addressed. I recommend its publication without further revisions.

Reviewer #2 (Remarks to the Author):

The authors have done a very good job in answering the questions raised by all the reviewers. I think that with the new experiments, the work is very detailed and will be important reading for the electrocatalysis community. I am left with a few minor comments:

- 1) I understand better what the authors are suggesting with the term "hazardous hydrogen", but I still have trouble understanding the term "deleterious organic hydrogen sources" with the acid formic example. Indeed, the protons come from H₂O in this work, but the acidity is controlled by adding KOH (in my opinion, the use of formic acid is no more deleterious than the latter).
- 2) l.248 "Notably, the -CH=CH₂ vibration peak of styrene (1630 cm⁻¹) was vanished, however, there is no shifts in the vibrations of C=C and C-H bonds of the benzene ring (1598 cm⁻¹ and 767 cm⁻¹) are observed on the surface of np-Cu, which suggests the interaction between the alkenyl group and np-Cu" This sentence is not clear to me. Are the authors suggesting that the disappearance of the C=C peak supports an interaction? If so, are there any examples in the literature to support this? However, I'm not sure what differentiates the two catalysts (np-Cu and Hnp-Cu₅₀Au₅₀) in this styrene case.

Responses to the Referees' Comments

Reviewer #2

The authors have done a very good job in answering the questions raised by all the reviewers. I think that with the new experiments, the work is very detailed and will be important reading for the electrocatalysis community. I am left with a few minor comments:

Response: Thank you for your positive comments on our manuscript. We have revised our manuscript accordingly.

Comment 1. I understand better what the authors are suggesting with the term "hazardous hydrogen", but I still have trouble understanding the term "deleterious organic hydrogen sources" with the acid formic example. Indeed, the protons come from H₂O in this work, but the acidity is controlled by adding KOH (in my opinion, the use of formic acid is no more deleterious than the latter).

Response: We sincerely appreciate your valuable comment. We are sorry for the loose of this description, following your comments, we revised it as following:

In this regard, a large amount of surface active hydrogen (H*) generated by water electrolysis can directly utilized as a sustainable hydrogen source for electrocatalytic transfer hydrogenation to avoid the use of hazardous hydrogen⁹.

Ref. [9] Liu, C., Wu, Y., Zhao, B. & Zhang, B. Designed nanomaterials for electrocatalytic organic hydrogenation using water as the hydrogen source. *Accounts. Chem. Res.* **56**, 1872-1883 (2023).

Comment 2. 248 "Notably, the -CH=CH₂ vibration peak of styrene (1630 cm⁻¹) was vanished, however, there is no shifts in the vibrations of C=C and C-H bonds of the benzene ring (1598 cm⁻¹ and 767 cm⁻¹) are observed on the surface of np-Cu, which suggests the interaction between the alkenyl group and np-Cu" This sentence is not clear to me. Are the authors suggesting that the disappearance of the C=C peak supports an interaction? If so, are there any examples in the literature to support this? However, I'm not sure what differentiates the two catalysts (np-Cu and Hnp-Cu₅₀Au₅₀) in this

styrene case.

Response: We sincerely thank you for these comments. According to the experimental results of electrocatalytic semi-hydrogenation of phenylacetylene, np-Cu would have the over-hydrogenation products leading to low selectivity, while Hnp-Cu₅₀Au₅₀ has excellent selectivity. Therefore, we used styrene as the substrate to investigate the highly selective source of Hnp-Cu₅₀Au₅₀ through in situ Raman spectroscopy. Styrene has three characteristic peaks, C=C (1598 cm⁻¹) and C-H (767 cm⁻¹) bonds belong to aromatic ring and the -CH=CH₂ (1630 cm⁻¹) belong to styrene. Under applied voltage, the C=C (1598 cm⁻¹) and C-H (767 cm⁻¹) bonds remain unchanged in the presence of np-Cu and Hnp-Cu₅₀Au₅₀, indicating the weak adsorption of the aromatic ring on the surface of catalysts. For the -CH=CH₂ (1630 cm⁻¹) bond, it still remain unchanged in the presence of Hnp-Cu₅₀Au₅₀, indicating a negligible interaction between the alkenyl group and Hnp-Cu₅₀Au₅₀, which can avoid its over-hydrogenation to form alkane, hence high alkene selectivity. However, the -CH=CH₂ (1630 cm⁻¹) bond was vanished in the presence of np-Cu, a similar phenomenon was observed in in situ Raman tests of electrocatalytic hydrogenation of alkynes (*J. Am. Chem. Soc.* **2022**, 144, 19456–19465, Figure S16), which suggests the interaction between the alkenyl group and np-Cu. In situ Raman results indicate that the alloying of Au with Cu can effectively weaken the adsorption of alkene on the catalyst surface to avoid over-hydrogenation into alkanes, thus boosting high alkene selectivity.

REVIEWERS' COMMENTS

Reviewer #2 (Remarks to the Author):

My last comments have been well addressed. I recommend publication without further revisions.

REVIEWER COMMENTS

Reviewer #1 (Remarks to the Author):

In this work, the authors reported a hierarchically nanoporous Cu₅₀Au₅₀ (Hnp-Cu₅₀Au₅₀) alloy for semi-hydrogenation of alkynes with high yield and a record Faradaic efficiency via the combining of alloying and confinement effect. The authors utilized comprehensive and impressive in-situ tools and finite element simulations to explore the intrinsic reasons of the outstanding performances, achieving a very convincing conclusion. The results are solid, and the conclusion can potentially extend to other similar systems for guiding future electrocatalysis design. I have some issues require authors to address and suggest revision for publication of this manuscript in Nature Communications.

Major questions:

1. The FT-EXAFS spectrum for Hnp-Cu₅₀Au₅₀ shows the much lower Cu-Au peak intensity in comparison with Cu foil (Fig 1g), I think the main reason is due to the grains become smaller.
2. The Hnp-Cu₅₀Au₅₀ exhibits a prominent peak at ~1.5 Å from the Cu-O bonds (Fig 1g). However, it disappeared during OCV testing (Fig 3b, Fig 3d). Please provide additional explanation.
3. The authors should check the grammar and writing standards in the manuscript.
4. The absorption edge positions of Cu are mislabeled in operando XANES (Fig 3a, Fig 3c) (see 10.1038/s41467-018-02819-7).
5. The authors need provide the full spectrum of XPS.
6. There should have a carbon peak in the XRD test results of the Hnp-Cu₅₀Au₅₀ after the reaction (Fig S13).

Reviewer #2 (Remarks to the Author):

This communication by Y. Tan and colleagues describe the synergistic effect of hierarchical nanopores and Cu-Au alloy for the electrocatalytic semi-hydrogenation of terminal alkynes. The nanoconfinement strategy achieves high efficiency (including high FE, which is sometimes a neglected parameter in this field) thanks to a higher local K⁺ concentration. The study is very well presented, nicely combining insights from various spectroscopic (including operando XAS) and electrochemical techniques as well as theoretical (DFT) results to support the catalytic performance, making it an important read for the electrocatalytic hydrogenation community. Therefore, I would recommend the publication after the following points have been addressed:

- 1) Regarding the scope of the reaction, the authors should clarify that it applies only to terminal alkynes, unless proven otherwise. In my opinion, this should even be stated in the title of the manuscript.
- 2) In the introduction, the authors contrast hydrogenation based on grey hydrogen (methane reforming) with ECH. I think it's important to recall that hydrogen can be produced by electrolysis of water, as this is a mature process, unlike ECH. In my opinion, the advantage of the latter lies in the economy of the process steps and its

energy-saving potential, rather than in defossilization.

3) l. 48: “deleterious organic solvent”. I see no reason why electrocatalytic hydrogenation (ECH) should preclude the use of organic solvents. In fact, they are often needed to solubilize organic substrates, and dioxane is used in this work which, in my opinion, is a “deleterious organic solvent” (see also point 11).

4) The reference electrode used in this study (Hg/HgO) is not standard and the potential values are therefore more difficult for the reader to grasp. I would recommend converting the potential to ferrocenium/ferrocene standard or silver-based reference electrodes, or at least giving the conversion constant relative to these couples/electrodes.

5) I find the positive effect of a higher K⁺ concentration on the system remarkable. However, I would suggest probing the contribution of ionic strength. For instance, another salt that does not contain K⁺, such as +1M nBu₄NCl + 1M KOH electrolyte, can be added (instead of KCl).

6) l. 232: “return to their initial states” I think the authors should clarify this statement, as the green and black curves do not overlap in either case (insets in figures 3a and 3c). For Raman analyses (Figure 3e), it is surprising that styrene adsorbed on np-Cu, and in particular the signals associated with the C=C bond, are not shifted relative to styrene alone, contrary to what is observed for alkynes [J. Am. Chem. Soc. 2022, 144, 19456]. Can the authors comment on this point?

7) It would also be interesting to perform the same Raman analyses in the presence of the substrate (i.e. phenylacetylene) to obtain further information on the binding of this species to the catalyst.

8) As constant-potential electrolysis has been carried out, it would be very interesting if the authors could estimate the overpotential applied in this work (at -1.3V, for instance). It is particularly important for estimating the energy efficiency of a system (beyond FE).

9) It would also be very useful, in my opinion, to have the over-reduction pathways computed by DFT (to complete figure 3f) to see if they can also support the high selectivity observed.

10) l.310 to 313: When describing the mechanism, it is not always clear whether the proposed steps take place in the inner or outer-sphere of the catalyst, particularly for radical intermediates, although it is probably difficult to distinguish for each of the steps.

11) l. 324: “environmental conditions” I don't find this statement very clear. Does it mean environmentally friendly? If so, this is far from being the case when using highly basic aqueous media combined with relatively toxic dioxane.

Also a few minor points:

12) The three terms Faraday, Faradic and Faradaic efficiencies are used throughout this manuscript and in the ESI. I suggest using only the last one.

13) l. 329: “alkynes” should be read as “alkenes”

Responses to the Referees' Comments

We thank the referees for their valuable comments and positive endorsement to our manuscript. We have carefully considered the referees' comments and revised the manuscript accordingly. Our responses and corresponding revisions are as follows:

Reviewer #1

In this work, the authors reported a hierarchically nanoporous Cu₅₀Au₅₀ (Hnp-Cu₅₀Au₅₀) alloy for semi-hydrogenation of alkynes with high yield and a record Faradaic efficiency via the combining of alloying and confinement effect. The authors utilized comprehensive and impressive in-situ tools and finite element simulations to explore the intrinsic reasons of the outstanding performances, achieving a very convincing conclusion. The results are solid, and the conclusion can potentially extend to other similar systems for guiding future electrocatalysis design. I have some issues require authors to address and suggest revision for publication of this manuscript in Nature Communications.

Response: Thank you for your positive comments on our manuscript. We have revised our manuscript accordingly.

Comment 1. The FT-EXAFS spectrum for Hnp-Cu₅₀Au₅₀ shows the much lower Cu-Au peak intensity in comparison with Cu foil (Fig 1g), I think the main reason is due to the grains become smaller.

Response: Thank you for your comments. Following the comment, we have carefully revised our manuscript and the details are listed below:

Importantly, the FT-EXAFS spectrum for Hnp-Cu₅₀Au₅₀ shows the much lower Cu-Au peak intensity in comparison with Cu foil, resulting from the grains become smaller, which could formation of abundant unsaturated coordination atoms and creation of defective surfaces.

Comment 2. The Hnp-Cu₅₀Au₅₀ exhibits a prominent peak at ~1.5 Å from the Cu-O

bonds (Fig 1g). However, it disappeared during OCV testing (Fig 3b, Fig 3d). Please provide additional explanation.

Response: We appreciate the questions from this reviewer. This is attributed to the surface oxide, which is removed by CV scanning before conducting the OCV testing.

Comment 3. The authors should check the grammar and writing standards in the manuscript.

Response: Many thanks for these comments. We have revised our manuscript accordingly.

Comment 4. The absorption edge positions of Cu are mislabeled in operando XANES (Fig 3a, Fig 3c) (see 10.1038/s41467-018-02819-7).

Response: We appreciate you for this insightful and constructive recommendation. Following your comments, we adjusted the labeled absorption edge positions of Cu and added the magnified rising edge XANES regions in the revised manuscript for the better comparison.

Figure R1. (a) Operando XANES spectra of Hnp-Cu₅₀Au₅₀ recorded at Cu K-edge (1 M KOH + phenylacetylene). (c) Operando XANES spectra of Hnp-Cu₅₀Au₅₀ recorded at Cu K-edge (1 M KOH + styrene).

Comment 5. The authors need provide the full spectrum of XPS.

Response: We appreciate the reviewer for this nice suggestion. Following this

comment, we have supplement the XPS full spectrum data as follows (see Figure S6 in revised supporting information).

Figure R2. XPS full spectra of Hnp-Cu₅₀Au₅₀.

Comment 6. There should have a carbon peak in the XRD test results of the Hnp-Cu₅₀Au₅₀ after the reaction.

Response: Thank you for the kind suggestion.

Figure R3. XRD patterns of Hnp-Cu₅₀Au₅₀ before and after reaction.

Reviewer #2

This communication by Y. Tan and colleagues describe the synergistic effect of hierarchical nanopores and Cu-Au alloy for the electrocatalytic semi-hydrogenation of terminal alkynes. The nanoconfinement strategy achieves high efficiency (including high FE, which is sometimes a neglected parameter in this field) thanks to a higher local K^+ concentration. The study is very well presented, nicely combining insights from various spectroscopic (including operando XAS) and electrochemical techniques as well as theoretical (DFT) results to support the catalytic performance, making it an important read for the electrocatalytic hydrogenation community. Therefore, I would recommend the publication after the following points have been addressed:

Response: Thank you for your positive comments on our manuscript. We have revised our manuscript accordingly.

Comment 1. Regarding the scope of the reaction, the authors should clarify that it applies only to terminal alkynes, unless proven otherwise. In my opinion, this should even be stated in the title of the manuscript.

Response: We sincerely appreciate your valuable comment. Following your comments, we have revised the title of the paper as follows:

“Coupled Alloying and Confinement effects on Hierarchically Nanoporous CuAu for Efficient Electrocatalytic Semi-Hydrogenation of Terminal Alkynes”

Comment 2. In the introduction, the authors contrast hydrogenation based on grey hydrogen (methane reforming) with ECH. I think it's important to recall that hydrogen can be produced by electrolysis of water, as this is a mature process, unlike ECH. In my opinion, the advantage of the latter lies in the economy of the process steps and its energy-saving potential, rather than in defossilization.

Response: We appreciate the suggestions from this reviewer. We sincerely appreciate the reviewer for this reminding. We have revised the manuscript as follows:

“However, a large part of the gaseous H_2 source of the TCH processes is produced by

the fossil fuel-based steam reforming process that inevitably results in high energy consumption and releases massive amounts of CO₂⁷. Alternatively, electrochemical hydrogenation technology, powered by renewable and clean energy, which would be more attractive and sustainable due to its low cost, high safety and environment friendly⁸.”

Ref. [7] Han, G., Li, G. & Sun, Y. Electrocatalytic dual hydrogenation of organic substrates with a Faradaic efficiency approaching 200%. *Nat. Catal.* **6**, 224-233 (2023).

Ref. [8] Tang, C., Zheng, Y., Jaroniec, M. & Qiao, S. Z. Electrocatalytic refinery for sustainable production of fuels and chemicals. *Angew. Chem. Int. Ed.* **60**, 19572-19590 (2021).

Comment 3. 48: “deleterious organic solvent”. I see no reason why electrocatalytic hydrogenation (ECH) should preclude the use of organic solvents. In fact, they are often needed to solubilize organic substrates, and dioxane is used in this work which, in my opinion, is a “deleterious organic solvent” (see also point 11).

Response: Thank you very much for this comment. We are sorry for the loose of this description. What we want to express is deleterious organic hydrogen sources (e.g., formic acid) causes several concerns regarding the safety and sustainability. And we have revised our manuscript as follows:

“In this regard, a large amount of surface active hydrogen (H*) generated by water electrolysis can directly utilized as a sustainable hydrogen source to electrocatalytic transfer hydrogenation, avoiding the use of hazardous hydrogen or deleterious organic hydrogen sources (e.g., formic acid)⁹.”

Ref. [9] Liu, C., Wu, Y., Zhao, B. & Zhang, B. Designed nanomaterials for electrocatalytic organic hydrogenation using water as the hydrogen source. *Accounts. Chem. Res.* **56**, 1872-1883 (2023).

Comment 4. The reference electrode used in this study (Hg/HgO) is not standard and the potential values are therefore more difficult for the reader to grasp. I would recommend converting the potential to ferrocenium/ferrocene standard or silver-based

reference electrodes, or at least giving the conversion constant relative to these couples/electrodes.

Response: We appreciate the reviewer for this nice suggestion. We have convert the potential to reversible hydrogen electrode (RHE) in manuscript and ESI by the following Nernst equation:

$$E(\text{vs RHE}) = E(\text{vs Hg/HgO}) + 0.098 \text{ V} + 0.0591 \times \text{pH}$$

1.0 M KOH (pH = 13.6)

Comment 5. I find the positive effect of a higher K⁺ concentration on the system remarkable. However, I would suggest probing the contribution of ionic strength. For instance, another salt that does not contain K⁺, such as +1M nBu₄NCl + 1M KOH electrolyte, can be added (instead of KCl).

Response: We appreciate you for this constructive recommendation. Following your insightful comments, we conducted a control experiment using nBu₄NCl instead of KCl added in 1 M KOH to investigate the effect of ionic strength for reaction performance. The experimental results indicate that the improvement in reaction performance is primarily due to the promotion of water dissociation by K⁺ rather than the contribution of ion strength (Figure R4). The results are added in Figure S11.

Figure R4. Conversions of phenylacetylene, selectivity and FEs of styrene over Hnp-Cu₅₀Au₅₀ in 1 M KOH + 1 M KCl and 1 M KOH + 1 M nBu₄NCl.

Comment 6. 232: “return to their initial states” I think the authors should clarify this statement, as the green and black curves do not overlap in either case (insets in figures 3a and 3c).

For Raman analyses (Figure 3e), it is surprising that styrene adsorbed on np-Cu, and in particular the signals associated with the C=C bond, are not shifted relative to styrene alone, contrary to what is observed for alkynes [J. Am. Chem. Soc. 2022, 144, 19456]. Can the authors comment on this point?

Response: We sincerely thank you for these comments. For your convenience, we provide a point-by-point response to these comments:

(1) We are sorry for the unclear description. Following this comment, we revised this description as following:

“Impressively, when the potential returns to OCV, the Cu valence state shift to the lower energies and the bond length of Cu-Au decreases, which can be persuasively ascribed to the rapid desorption process of reactants or reaction intermediates on the catalyst surface. ”

(2) We are sorry for the wrong description in this part. Following these comments, we retested the in situ Raman (Figure R5) and revised this part as following:

“Notably, the -CH=CH₂ vibration peak of styrene (1630 cm⁻¹) was vanished, however, there is no shifts in the vibrations of C=C and C-H bonds of the benzene ring (1598 cm⁻¹ and 767 cm⁻¹) are observed on the surface of np-Cu, which suggests the interaction between the alkenyl group and np-Cu. In contrast, no clear shifts of these characteristic peaks with Hnp-Cu₅₀Au₅₀ (Figure 3e). These results indicate that the alloying of Au with Cu can effectively weaken the adsorption of alkene on the catalyst surface to avoid over-hydrogenation into alkanes, thus boosting high alkene selectivity^{20,41}. ”

Figure R5. In situ Raman tests for electrocatalytic hydrogenation of styrene over np-Cu and Hnp-Cu₅₀Au₅₀ at -0.4 V vs. RHE.

Ref. [20] Bai, L. et al. Efficient industrial-current-density acetylene to polymer-grade ethylene via hydrogen-localization transfer over fluorine-modified copper. *Nat. Commun.* **14**, 8384 (2023).

Ref. [41] Li, H. et al. σ -alkynyl adsorption enables electrocatalytic semihydrogenation of terminal alkynes with easy-reducible/passivated groups over amorphous PdS_x nanocapsules. *J. Am. Chem. Soc.* **144**, 19456-19465 (2022).

Comment 7. It would also be interesting to perform the same Raman analyses in the presence of the substrate (i.e. phenylacetylene) to obtain further information on the binding of this species to the catalyst.

Response: We sincerely thank you for this nice suggestion. Following the comment, we performed the in situ Raman analyses in the presence of the phenylacetylene (Figure R6) and supplement the information in our manuscript (Figure S18) as below:

“The C \equiv C stretching vibration peak of phenylacetylene is blue-shifted from 2108 to 2201 and 2218 cm⁻¹ and vanishing of the terminal C-H vibration peaks in the presence of np-Cu and Hnp-Cu₅₀Au₅₀, which suggests the interaction between the alkynyl group and catalysts. Additionally, the vibrations of the C=C skeleton (1598 cm⁻¹) and C-H (1090 cm⁻¹ for the in-plane rocking mode and 787 cm⁻¹ for the out-of-plane bending)

bonds of the benzene ring remain unchanged, indicating a negligible interaction with np-Cu and Hnp-Cu₅₀Au₅₀. Thus, phenylacetylene adsorbs on the np-Cu and Hnp-Cu₅₀Au₅₀ with only σ -alkynyl bonding adsorption mode⁴¹. "

Figure R6. In situ Raman tests in a mixed 1.0 M KOH/Diox solution for electrocatalytic hydrogenation of phenylacetylene over np-Cu and Hnp-Cu₅₀Au₅₀ at -0.4 V vs. RHE.

Ref. [41] Li, H. et al. σ -Alkynyl Adsorption Enables Electrocatalytic Semihydrogenation of Terminal Alkynes with Easy-Reducible/Passivated Groups over Amorphous PdSx Nanocapsules. *J. Am. Chem. Soc.* **144**, 19456–19465 (2022).

Comment 8. As constant-potential electrolysis has been carried out, it would be very interesting if the authors could estimate the overpotential applied in this work (at -1.3 V, for instance). It is particularly important for estimating the energy efficiency of a system (beyond FE).

Response: Thank you for the kind suggestion.

The energy efficiency was defined as the ratio of fuel energy to applied electrical power, which was calculated by:

$$EE_{styrene} = ((E_{OER}^{\theta} - E_{styrene}^{\theta}) \times FE_{styrene}) / (E_{OER} - E_{styrene})$$

where $E_{styrene}^{\theta}$ represents the equilibrium potential of phenylacetylene electroreduction to styrene, which is calculated by DFT (0.49 V vs. RHE) (Figure R7 and Figure S26)^[6], E_{OER}^{θ} is the equilibrium potential of the oxygen evolution reaction (OER) (1.23 V vs. RHE), $FE_{styrene}$ is the Faradaic efficiency for styrene, and E_{OER} and $E_{styrene}$ are the

applied potentials.

According to the above formula, the energy efficiency is calculated, and we have added the following sentences to our manuscript:

Moreover, the energy efficiency (EE) of the Hnp-Cu₅₀Au₅₀ for semi-hydrogenation of phenylacetylene is further evaluated, which shows a high EE of 42% at -0.4 V vs. RHE (Figure R8 and Figure S14).

Figure R7. (a) Standard Gibbs free energies of H₂(g), C₈H₆, C₈H₈ and H₂O(l). (b) The standard equilibrium potential of phenylacetylene-to-styrene conversion routes.

Figure R8. The energy efficiency of Hnp-Cu₅₀Au₅₀ for semi-hydrogenation of phenylacetylene at different applied potentials.

Ref. [6] Shi, R. et al. Room-temperature electrochemical acetylene reduction to ethylene with high conversion and selectivity. *Nat. Catal.* **4**, 565–574 (2021).

Comment 9. It would also be very useful, in my opinion, to have the over-reduction pathways computed by DFT (to complete figure 3f) to see if they can also support the

high selectivity observed.

Response: We are grateful to the reviewer for this nice comment. We have completed the over-reduction pathway of phenylacetylene hydrogenation through DFT. For the pathway from styrene to phenylethane ($*C_8H_8 \rightarrow *C_8H_9 \rightarrow *C_8H_{10}$), we found that the barrier for hydrogenation of styrene is smaller than the energy of styrene required for desorption into free molecules. We speculate that this is because our actual catalyst is an alloy phase metal of Cu and Au, and the atomic occupancy of its FCC structure has a certain degree of disorder. It is difficult to construct a matching model structure through DFT calculations, and the computational workload is significant. We will perform more comprehensive and detailed calculations in the follow-up work. Many thanks for your understanding. However, our current results (Figure R9 and Figure S19), are sufficient to explain the excellent hydrogenation performance of $Cu_{50}Au_{50}$ from phenylacetylene to styrene. The Cu (111) surface exhibits strong adsorption of C_8H_x species, making it easier to hydrogenate to phenylethane. However, the addition of Au atoms weakens the adsorption effect on the Cu surface, which can regulate a moderate surface of hydrogenation to the alkene. Meanwhile, the provision of H protons on the Au (111) surface is more difficult, while the introduction of Cu can provide a moderate hydrogen source, which can also prevent over hydrogenation to phenylethane.

Figure R9. Gibbs free energy diagram for alkyne semi-hydrogenation reactions over Cu (111), Au (111), and $Cu_{50}Au_{50}$.

Comment 10. 310 to 313: When describing the mechanism, it is not always clear whether the proposed steps take place in the inner or outer-sphere of the catalyst, particularly for radical intermediates, although it is probably difficult to distinguish for each of the steps.

Response: We thank the reviewer for presenting this nice question.

For outer-sphere reactions, the substrates, intermediates, and products do not interact strongly with the electrode material and electron transfer occurs by tunnelling across a solvation layer, while in an inner-sphere reaction there is a strong interaction between the substrate and the electrode surface^[1, 2]. According to the results of in situ Raman and operando XAS, there is a strong interaction between phenylacetylene and the catalyst. Therefore, the reaction steps take place in the inner-sphere of the catalyst. Following this comment, we revised our manuscript as follows:

Then, the H* reaction with the C≡C bond of a nearby phenylacetylene to form the carbon radical intermediates in the inner-sphere of the catalyst, and immediately couple with another H* to produce the styrene products. Subsequently, desorption of styrene on the alloy surface regenerates the catalytic sites for the next reaction cycle

Ref. [1] Zhou, Hua., Li, Z., Ma, L. & Duan, H. Electrocatalytic oxidative upgrading of biomass platform chemicals: from the aspect of reaction mechanism. *Chem. Commun.* **58**, 897 (2022).

Ref. [2] Zhang, P. & Sun, L. Electrocatalytic hydrogenation and oxidation in aqueous conditions. *Chin. J. Chem.* **38**, 996—1004 (2020).

Comment 11. 324: “environmental conditions” I don't find this statement very clear. Does it mean environmentally friendly? If so, this is far from being the case when using highly basic aqueous media combined with relatively toxic dioxane.

Response: We appreciate you for this valuable recommendation. We are sorry for the loose of this description. Compared with the high temperature and pressure conditions of thermochemical hydrogenation, electrochemical hydrogenation was carried out at room temperature and atmospheric pressure. What we want to express is that the reaction conditions are mild.

Comment 12. The three terms Faraday, Faradic and Faradaic efficiencies are used throughout this manuscript and in the ESI. I suggest using only the last one.

Response: We appreciate you for this valuable recommendation. We have made revisions in the manuscript and ESI.

Comment 13. 329: “alkynes” should be read as “alkenes”

Response: We really appreciate your reminder. Following your comments, we revised it as following:

By virtue of operando XAS, in situ Raman measurements, and DFT calculations, we confirmed that the incorporation of Au atoms into the Cu matrix could effectively optimize the electronic structure of alloys, which can suppress the HER and weaken adsorption of alkenes to inhibit over-hydrogenation, thus resulting in a superior semi-hydrogenation selectivity to targeted alkenes.

Reviewer #1 (Remarks to the Author):

My comments have been well addressed. I recommend its publication without further revisions.

Reviewer #2 (Remarks to the Author):

The authors have done a very good job in answering the questions raised by all the reviewers. I think that with the new experiments, the work is very detailed and will be important reading for the electrocatalysis community. I am left with a few minor comments:

1) I understand better what the authors are suggesting with the term "hazardous hydrogen", but I still have trouble understanding the term "deleterious organic hydrogen sources" with the acid formic example. Indeed, the protons come from H₂O in this work, but the acidity is controlled by adding KOH (in my opinion, the use of formic acid is no more deleterious than the latter).

2) 1.248 "Notably, the -CH=CH₂ vibration peak of styrene (1630 cm⁻¹) was vanished, however, there is no shifts in the vibrations of C=C and C-H bonds of the benzene ring (1598 cm⁻¹ and 767 cm⁻¹) are observed on the surface of np-Cu, which suggests the interaction between the alkenyl group and np-Cu" This sentence is not clear to me. Are the authors suggesting that the disappearance of the C=C peak supports an interaction? If so, are there any examples in the literature to support this? However, I'm not sure what differentiates the two catalysts (np-Cu and Hnp-Cu₅₀Au₅₀) in this styrene case.

Responses to the Referees' Comments

Reviewer #2

The authors have done a very good job in answering the questions raised by all the reviewers. I think that with the new experiments, the work is very detailed and will be important reading for the electrocatalysis community. I am left with a few minor comments:

Response: Thank you for your positive comments on our manuscript. We have revised our manuscript accordingly.

Comment 1. I understand better what the authors are suggesting with the term "hazardous hydrogen", but I still have trouble understanding the term "deleterious organic hydrogen sources" with the acid formic example. Indeed, the protons come from H₂O in this work, but the acidity is controlled by adding KOH (in my opinion, the use of formic acid is no more deleterious than the latter).

Response: We sincerely appreciate your valuable comment. We are sorry for the loose of this description, following your comments, we revised it as following:

In this regard, a large amount of surface active hydrogen (H*) generated by water electrolysis can directly utilized as a sustainable hydrogen source for electrocatalytic transfer hydrogenation to avoid the use of hazardous hydrogen⁹.

Ref. [9] Liu, C., Wu, Y., Zhao, B. & Zhang, B. Designed nanomaterials for electrocatalytic organic hydrogenation using water as the hydrogen source. *Accounts. Chem. Res.* **56**, 1872-1883 (2023).

Comment 2. 248 "Notably, the -CH=CH₂ vibration peak of styrene (1630 cm⁻¹) was vanished, however, there is no shifts in the vibrations of C=C and C-H bonds of the benzene ring (1598 cm⁻¹ and 767 cm⁻¹) are observed on the surface of np-Cu, which suggests the interaction between the alkenyl group and np-Cu" This sentence is not clear to me. Are the authors suggesting that the disappearance of the C=C peak supports an interaction? If so, are there any examples in the literature to support this? However, I'm not sure what differentiates the two catalysts (np-Cu and Hnp-Cu₅₀Au₅₀) in this

styrene case.

Response: We sincerely thank you for these comments. According to the experimental results of electrocatalytic semi-hydrogenation of phenylacetylene, np-Cu would have the over-hydrogenation products leading to low selectivity, while Hnp-Cu₅₀Au₅₀ has excellent selectivity. Therefore, we used styrene as the substrate to investigate the highly selective source of Hnp-Cu₅₀Au₅₀ through in situ Raman spectroscopy. Styrene has three characteristic peaks, C=C (1598 cm⁻¹) and C-H (767 cm⁻¹) bonds belong to aromatic ring and the -CH=CH₂ (1630 cm⁻¹) belong to styrene. Under applied voltage, the C=C (1598 cm⁻¹) and C-H (767 cm⁻¹) bonds remain unchanged in the presence of np-Cu and Hnp-Cu₅₀Au₅₀, indicating the weak adsorption of the aromatic ring on the surface of catalysts. For the -CH=CH₂ (1630 cm⁻¹) bond, it still remain unchanged in the presence of Hnp-Cu₅₀Au₅₀, indicating a negligible interaction between the alkenyl group and Hnp-Cu₅₀Au₅₀, which can avoid its over-hydrogenation to form alkane, hence high alkene selectivity. However, the -CH=CH₂ (1630 cm⁻¹) bond was vanished in the presence of np-Cu, a similar phenomenon was observed in in situ Raman tests of electrocatalytic hydrogenation of alkynes (*J. Am. Chem. Soc.* **2022**, 144, 19456–19465, Figure S16), which suggests the interaction between the alkenyl group and np-Cu. In situ Raman results indicate that the alloying of Au with Cu can effectively weaken the adsorption of alkene on the catalyst surface to avoid over-hydrogenation into alkanes, thus boosting high alkene selectivity.